# On the Consistency of Scale Among Experiments, Theory, and Simulation

James E. McClure[1], Amanda L. Dye[2], Cass T. Miller[2], and William G. Gray[3]

[1]Advanced Research Computing, Virginia Tech, Blacksburg, Virginia 24601-0123, USA
[2]Department of Environmental Sciences and Engineering, University of North Carolina, Chapel Hill, North Carolina 27599-7431
[3]Curriculum for the Environment and Ecology, University of North Carolina, Chapel Hill, North Carolina 27599-3135

*Correspondence to:* William G. Gray (GrayWG@unc.edu)

**Abstract.** As a tool for addressing problems of scale, we consider an evolving approach known as the thermodynamically constrained averaging theory (TCAT) which has broad applicability to hydrology. We consider the case of modeling of two-fluid-phase flow in porous media, and we focus on issues of scale as they relate to various measures of pressure, capillary pressure, and state equations needed to produce solvable models. We apply TCAT to perform physics-based data assimilation to understand how the internal behavior influences the macroscale state of two-fluid porous medium systems. A microfluidic experimental method and a lattice Boltzmann simulation method are used to examine a key deficiency associated with standard approaches. In a hydrologic process such as evaporation, the water content will ultimately be reduced below the so-called irreducible wetting phase saturation determined from experiments. This is problematic since the derived closure relationships cannot predict the associated capillary pressures for these states. We demonstrate that the irreducible wetting-phase saturation is an artifact of the experimental design, caused by the fact that the boundary pressure difference does not approximate the true capillary pressure. Using averaging methods, we compute the true capillary pressure for fluid configurations at and below the irreducible wetting phase saturation. Results of our analysis include a state function for the capillary pressure expressed as a function of fluid saturation and interfacial area.

## 1 Introduction

Hydrologic systems are typically investigated using some combination of experimental, computational, and theoretical approaches. Each of these classes of approaches has played a central role in advancing knowledge. The years spanning the career of Eric F. Wood have witnessed a remarkable development in the ability to study experimentally the elements that comprise the hydrologic universe. The subsurface is a porous medium system that receives experimental attention designed to identify the small-scale fluid distributions within the solid matrix, intermediate scale behavior through laboratory study, and also the response of an aquifer to imposed forces (e.g., Wildenschild and Sheppard, 2013; Dye et al., 2015; Alizadeh and Piri, 2014; Knödel et al., 2007). Turbulence in surface flows and its impact in rivers, estuaries, and oceans for flow, sediment transport, and dissolved species transport is examined using a broad range of experimental techniques (e.g., Bradshaw, 1971; Chanson, 2009; D'Asaro, 2014; Bernard and Wallace, 2002). Atmospheric experiments designed to support theoretical models of turbulence,

typically using lidar systems, and to gain insight into turbulence structures have also generated large quantities of data (Sathe and Mann, 2013; Collins et al., 2015; Fuentes et al., 2014). Other studies involve examination of snow pack, desertification, and changes in land usage (Deems et al., 2013; Hermann and Sop, 2016; Lillesand et al., 2015; Nickerson et al., 2013).

Complementing the advancing ability of experimental study is the development of simulation tools for various aspects of
hydrologic systems that make use of advanced computer technology (e.g., Miller et al., 2013; Flint et al., 2013; Kauffeldt et al., 2016; Paiva et al., 2011; Dietrich et al., 2013; Zhou and Li, 2011; Miller et al., 1998; Bauer et al., 2015; Dudhia, 2014). These models of watersheds, rivers and estuaries, and subsurface regions usually make use of traditional equations with the advances occurring through the ability of modern computer architecture to handle larger problems using parallel computing and more elegant, efficient graphical user interfaces.

A third element of advancing modeling of water resources systems is the development of theory that accounts for physical processes. On one hand, forming theoretical advances for mechanistic models based upon conservation equations can be viewed as the standard challenges of accounting more completely for conserved quantities and of developing closure relations for dissipative processes. However, the need to pose closure relations at scales that are consistent with those at which the problems have been formulated creates a need for a variety of constitutive proposals. Furthermore, consistency of models
requires that equation formulations be consistent across scales such that variables developed at a smaller scale can inform the equations employed at a larger scale. Overall, these considerations lead to identifying scale and scaling behavior in both time and space as important challenges in posing models (Wood, 1995; Wang et al., 2006; Skøien et al., 2003; Pechlivanidis et al., 2011; Gleeson and Paszkowski, 2014; Gentine et al., 2012; Blöschl, 2001).

In an era of unprecedented data generation, opportunities to use multiscale averaging theory to develop physics-based data
assimilation strategies based have never been more evident. The challenge of performing meaningful theoretical, experimental, and computational analyses is constrained by the need to ensure that the length and time scales of quantities arising in each approach can be related. The scales of experimental data, variables appearing in equations, and computed quantities must be the same if they are to be compared in any meaningful way. As a prerequisite for this to happen, data generated by any of the methods must be consistent across the range of scales considered (Ly et al., 2013; Kauffeldt et al., 2013).

While the desire for consistencies across scales and approaches is conceptually simple to understand, it has proven to be a difficult practical objective to meet. The change in scale of conservation and balance equations can be accomplished rather easily. The problem with applying these equations lies in the aforementioned need to average some intensive variables, the requirement that closure conditions be proposed at the larger scale, and the need to account for the dynamics of new quantities that arise in the change of scale. Without accounting for all of these items properly, models are doomed to fail. An
essential element in ensuring success is the averaging of thermodynamic relations to the larger scale (Gray and Miller, 2013). This provides linkage of variables across scales and also ensures that all physical processes are properly accounted for. For modeling rainfall-runoff processes, Wood et al. (1988) proposed the use of a representative elementary area as a portion of a watershed over which averaging can occur to develop a model. This idea was extended and applied by Blöschl et al. (1995). Subsequently, Reggiani et al. (1998) proposed treating a hydrologic system as a collection of interconnected lumped elements.
The lumping was accomplished by integration over individual portions of the system with distinct properties, e.g., aquifers,

streams, channels. This effort did not include integration of thermodynamic relations, and as a result did not properly account for the impact of gravitational potential in driving flow between system elements. An effort to address this shortcoming by a somewhat ad hoc introduction of gravitational forces (Reggiani et al., 1999) was only partially successful. Averaging of thermodynamic relations to lumped elements has since been presented (Gray and Miller, 2009).

Challenges in assuring consistency across scales have also been confronted in the modeling of porous medium systems. Special challenges have been encountered for two-fluid-phase flow, where upscaling leads to the introduction of quantities such as specific interfacial area, which is the area where two phases meet normalized by the volume of the region, and specific common curve length, which is length of a curve where three phases meet normalized by the volume of the region. Modeling of multiscale porous medium systems must also employ thermodynamics that is scale-consistent and included naturally as a part of the process. Because of the inability to overcome these challenges, most efforts to model multiscale, multiphase porous medium systems do not have thermodynamic constraints and full-scale consistency that would be expected in mature models. The thermodynamically constrained averaging theory (TCAT) approach is relatively refined and provides means to model systems that are inherently multiscale in nature and also to link disparate length scales, while representing the essential physics naturally and hierarchically with varying levels of sophistication. However, realizing these scale-consistent attributes requires new approaches, new equations of state, novel parameterizations, and, as with any new model, evaluation and validation.

## 2 Objectives

The overall goal of this work is to examine the impact of phase connectivity on scale consistency for two-fluid-phase porous medium systems. From the mathematical standpoint, the microscale and macroscale must provide a consistent view of the physics. In our approach, macroscale variables (such as phase pressures and capillary pressure) are explicitly defined in terms of microscale quantities to ensure that physical consistency is achieved. The resultant rigorous connection between the microscale and the macroscale can be exploited to understand and characterize how phase connectivity influences key macroscale quantities. In other words, we ensure consistency between information at small and large scales by using precise mathematics to change the scale of variables; and we also ensure that variables denoted as pertaining to theory, experiment, or simulation are defined such that they refer to quantities defined at the same scale and are directly comparable. The specific objectives of this work are:

- to formulate explicitly related microscale and macroscale descriptions of state variables important for traditional and evolving descriptions of capillary pressure;

- to determine state variables for capillary pressure using complementary experimental and computational approaches;

- to compare a traditional state equation approximation approach with a carefully formulated approach based in multiscale TCAT theory;

- to demonstrate the limitations of traditional state equation approaches for macroscale capillary pressure; and

– to examine the uniqueness of alternative state equation formulations for capillary pressure.

The objectives of this work are focused on a specific aspect of approaches commonly used to represent the behavior of porous medium systems. The physical size of the systems considered experimentally and computationally are small and idealized compared to field-scale hydrologic systems that motivate this work and which hydrologists collectively endeavor to better understand and describe quantitatively with higher fidelity than current approaches. Our hope in examining these fundamental issues is to advance basic understanding of hydrologic systems and to stimulate future work that might allow such advancements to be reduced to improved tools for practice in due course.

## 3 Background

Two spatial scales are of primary interest for the porous medium problems of focus herein: the microscale, which is often referred to as the pore scale; and the macroscale, which is often referred to as the porous medium continuum scale. At the microscale, the geometry of all phase distributions are fully resolved in space and in time, which makes it possible to locate interfaces where two phases meet and common curves where three phases meet. The equations governing the conservation of mass, momentum, and energy, the balance of entropy, and equilibrium thermodynamic relations are well established at the microscale. Microscale experimental work and modeling are active areas of research because of their relevance to understanding operative processes in complex porous medium systems that were previously impossible to observe. The macroscale is a scale for which a point is associated with some averaged properties of an averaging region comprising all phases, interfaces, and common curves present in the system. Notions such as volume fraction and specific interfacial area arise when a system is represented at the macroscale in terms of averaged measures of the state of the system. These additional measures are quantities that must be determined in the model solution process. Because of historical limitations on both computational and observational data, the macroscale has been the traditional scale at which models of natural porous media systems have been formulated and solved Closure relations at this scale are needed to yield well-posed models. Traditionally, these closure relations have been posited empirically and parameter estimation has been accomplished based upon relatively simple laboratory experiments. In general, traditional macroscale models, while the dominant class of model, suffer from several limitations related to the way in which such models are formulated and closed (Gray and Miller, 2014). A precise coupling between these disparate length scales has usually been ignored.

As efforts to model and link hydrologic elements in models advance, the ability to address scales effectively will become essential. For porous media, methods such as averaging, mixture theory, percolation theory, and homogenization have been employed to transform governing system equations from smaller to larger length scales (Hornung, 1997; Panfilov, 2000; Cushman, 1997). The goal of such approaches is to transform small-scale data to a larger scale such that it can be used to inform models that have been obtained by consistent transformation of conservation and balance equations across scales.

Averaging procedures have been used for analysis of porous media for approximately 50 years (e.g., Bear, 1972; Anderson and Jackson, 1967; Whitaker, 1986, 1999; Marle, 1967). The methods of averaging can be applied to single-fluid-phase systems as well as to multiphase systems. Success in the development of averaging equations for single-fluid-phase porous media

to obtain equations such as Darcy's Law has been achieved (e.g., Bachmat and Bear, 1964; Whitaker, 1967; Gray and O'Neill, 1976). These instances did not so much derive a flow equation as show that a commonly used flow equation could be obtained using averaging theorems and appropriate assumptions. Thus, these early efforts did not contribute significantly to objective development of flow equations that seek to capture important physical processes. They do serve to provide a systematic frame-work for developing larger scale equations. Work for two or more fluid phases in porous media has proven to be more difficult and has not been as illuminating.

The problems associated with trying to model multiple fluid phases in porous media include: (1) difficulties in properly accounting for interface properties, (2) lack of definition of macroscale intensive thermodynamic variables, (3) failure to account for system kinematics, and (4) challenges representing other important physical phenomena explicitly, such as contact angles and common curve behavior. These four difficulties sometimes impact the system description in combination.

Multiple-fluid-phase porous media differ from a single-fluid-phase porous medium system by the presence of the interface between the fluids. This interface is different from a fluid-solid interface because of its dynamics. The total amount of solid surface is roughly constant, or is slowly varying, for most natural solid materials. The fluid-fluid specific interfacial area changes in response to flow in the system and redistribution of phases. The time scale of this change is between that of the pore diameter divided by flow velocity and that of pore diameter divided by solid phase movement. These specific interfacial areas are important for their extent, surface tension, and curvature. They are the location where capillary forces are present. Thus, a physically consistent model must account for mass, momentum, and energy conservation at the interfaces; a model concerned only with phase behavior cannot represent capillary pressure in a mechanistically high-fidelity fashion (Gray et al., 2015). This shortcoming is evidenced, in part, by multi-valuedness when capillary pressure is proposed to be a function only of saturation (Albers, 2014).

Intensive variables that are introduced at the macroscale without consideration of microscale precursor values are also poorly defined. For example, a range of procedures for averaging microscale temperature can be employed that will lead to different macroscale values unless the microscale temperature is constant over the averaging region. Thus, mere speculation that a macroscale value exists fails to identify how or if this value is related to unique microscale variables and most certainly does not relate the macroscale variable to microscale quantities. The absence of a theoretical relation makes it impossible to reliably relate microscale measurements to larger scale representations (Essex et al., 2007; Maugin, 1999). Further confusion arises when pressure is proposed directly at the macroscale. Microscale capillary pressure is related to the curvature of the interface between fluid phases and does not depend on the pressures in the two phases themselves. At equilibrium, microscale capillary pressure becomes equal to the difference between phase pressures at the interface. Proposed representations of macroscale capillary pressure often specify that the capillary pressure is equal to the difference in some directly presumed quantities known as macroscale pressures of phases. These representations ignore both interface curvature and the fact that only when evaluated at the interface is the phase pressure useful for describing equilibrium capillary pressure. This is especially problematic when boundary pressures in an experimental cell are used to compute a so called "capillary pressure." Note that under these common experimental conditions, regions of entrapped non-wetting phase are not in contact with the non-wetting fluid that is observed on the boundary of the system.

The importance of kinematics is recognized, at least implicitly, in modeling many systems at reduced dimensionality or when averaging over a region the system occupies. For example, in the derivation of vertically integrated shallow water flow equations, a kinematic condition on the top surface is imposed based on the condition that no fluid crosses that surface (Vreugdenhil, 1995). Macroscale kinematic equations for interfaces between fluids in the absence of porous media have been proposed

in the context of boiling (Kocamustafaogullari and Ishii, 1995; Ishii et al., 2005). Despite the fact that interface reconfiguration has an important role in determining the properties and behavior of a multifluid porous medium system, attention to this feature is extremely limited (Gray and Miller, 2013; Gray et al., 2015). In some cases, models of two-fluid-phase flow in porous media have been proposed that do not account for either system kinematics or for interfacial stress (e.g. Niessner et al., 2011). Both are necessary components of physically realistic, high fidelity models.

The mixed success in posing appropriate theoretical models, making use of relevant data, and harnessing effective computer power to advance understanding of hydrologic systems is attributable to the inherent difficulty of each of these scientific activities. For progress to be made in enhancing understanding, a significant hurdle must be navigated that requires consistency among these three approaches and within each approach individually. We have found that by performing complementary microscale experimental and computational studies, we have formed a basis for being able to upscale data spatially with

insights into the operative time scales for the system (Gray et al., 2015). The small-scale data supports our quest for larger scale closure relations and eliminates confusion about concepts such as capillary pressure as a state function and dynamic processes that cause changes in the value of capillary pressure. Key to being able to develop faithful models are consistent scale change of thermodynamic relations and implementation of appropriate kinematic relations. The approach of combining sound theory, modern experimental techniques, and advanced computational techniques to the study of environmental systems

has applicability not only for the porous media systems emphasized here but also for large scale systems with interacting atmospheric, surface, and subsurface elements.

## 4   Theory

An important aspect of the issues of concern in this work is related to the various ways in which capillary pressure can be measured and the consequences of using traditional approaches that observe fluid pressures on the boundary of an experimental

cell and approximate the capillary pressure based upon the difference between the non-wetting phase pressure and the wetting phase pressure. However, even alternative approaches such as those based upon measurements using microtensiometers cannot resolve the issues of concern identified in this work. The differences among approaches are important, and commonly used approaches are flawed. In the formulation that follows, we show how microscale pressures can be averaged in a variety of ways as well as the relationship of these averaged pressures to the true capillary pressure. We note that averaging of pressures is

inherent in the formulation of macroscale models; and indeed measurement devices themselves provide averages over a length scale depending upon the device. The issues related to averaging cannot be avoided.

Averaging of any intensive variable (e.g., pressure, temperature, chemical potential) is problematic because there is no unique averaging procedure that can be employed. This is in contrast to obtaining an upscaled value of mass per volume by

integrating the microscale density over a volume to obtain the total mass and then dividing by the volume to get the upscaled density. Pressure, for example, is a force per area or, alternatively, an energy per volume. Averaging pressure over some area in a region as opposed to averaging over the volume of the region can give different values. Thus, it is imperative to identify pressure averages in ways that they arise in equations and in data collection. Correct identification of an averaged pressure and

association of that average with a particular process or element of an equation is essential if the physics of a system are going to be described well at the macroscale. For this reason, we carefully define the larger scale variables that will be used in analyzing the simulated system and describing system physics in this section. We also highlight the importance of identifying capillary pressure as an intrinsic property of an interface rather than as having an identity that is based on properties of juxtaposed phases.

Direct upscaling can be performed based on microscale information, providing an opportunity to explore aspects of macroscale system behavior that have previously been overlooked. Underpinning this exploration is the precise definition of macroscale quantities. TCAT models are derived from first-principles starting from the microscale. At the macroscale, important quantities such as phase pressures, specific interfacial areas, curvatures, and other averaged quantities are defined unambiguously based on the microscale state (e.g. Gray and Miller, 2014). For two-fluid-phase flow we consider the wetting phase ($w$), the non-wetting

phase ($n$), and the solid phase ($s$) within a domain $\Omega$. Each phase occupies part of the domain, $\Omega_\alpha$, where $\alpha = \{w, n, s\}$. The intersection between any two phases is an interface. The three interfaces are denoted by $\Omega_{wn}$ $\Omega_{ws}$, and $\Omega_{ns}$. Finally, the common curve $\Omega_{wns}$ is defined by the juncture of all three phases. The TCAT two-phase model is developed based on averaging with the complete set of entities, with the index set $\mathcal{J} = \{w, n, s, wn, ws, ns, wns\} = \mathcal{J}_P \cup \mathcal{J}_I \cup \mathcal{J}_C$ chosen to include all three phases $\mathcal{J}_P = \{w, n, s\}$, the interfaces $\mathcal{J}_I = \{wn, ws, ns\}$, and the common curve $\mathcal{J}_C = \{wns\}$. Based on this, the pore space

is defined as the union of the domains for the two fluids $\mathcal{D}_f = \Omega_w \cup \Omega_n$.

  Macroscale quantities can be determined explicitly from microscale information based on averages. In this work, the form for averages is

$$\left\langle P \right\rangle_{\alpha,\beta} = \frac{\int_{\Omega_\alpha} P d\mathbf{r}}{\int_{\Omega_\beta} d\mathbf{r}} \, , \tag{1}$$

where $P$ is the microscale quantity being averaged. The domains for integration can be the full domain $\Omega$, the entity domains

$\Omega_\alpha$ for $\alpha \in \mathcal{J}$, or their boundary $\Gamma_\alpha$. The boundary of an entity can be further sub-divided into an internal component $\Gamma_{\alpha i}$ and an external component $\Gamma_{\alpha e}$, which together yield $\Gamma_\alpha = \Gamma_{\alpha i} \cup \Gamma_{\alpha e}$. The external boundary is simply $\Gamma_{\alpha e} = \Omega_\alpha \cap \Gamma$.

  The volume fractions, specific interfacial areas, and specific common curve length are each extent measures that can be formulated as

$$\epsilon^{\overline{\overline{\alpha}}} = \left\langle 1 \right\rangle_{\Omega_\alpha, \Omega} \, . \tag{2}$$

The volume fractions correspond to $\alpha \in \mathcal{J}_P$; specific interfacial areas correspond to averaging over a two-dimensional interface for $\alpha \in \mathcal{J}_I$; and the specific common curve length corresponds to averaging over a one-dimensional common curve for $\alpha = wns$. The system porosity, $\epsilon$, is directly related to the solid phase volume fraction by

$$\epsilon = 1 - \epsilon^{\overline{\overline{s}}} \, . \tag{3}$$

The wetting phase saturation, $s^{\overline{\overline{w}}}$, can also be expressed in terms of the extent measures,

$$s^{\overline{\overline{w}}} = \frac{\epsilon^{\overline{\overline{w}}}}{1 - \epsilon^{\overline{\overline{s}}}} = \frac{\epsilon^{\overline{\overline{w}}}}{\epsilon} \; . \tag{4}$$

At the macroscale, various averages arise for the fluid pressures. For flow processes, the relevant quantity is an intrinsic average of the microscale fluid pressure, $p_\alpha$, expressed as

$$p^\alpha = \left\langle p_\alpha \right\rangle_{\Omega_\alpha, \Omega_\alpha} \tag{5}$$

for $\alpha \in \mathcal{J}_f$, which is the index set of fluid phases. In most laboratory experiments phase pressures are measured at the boundary. Pressure transducers can be placed within a domain at pre-selected locations, which still does not provide a dense, non-intrusive measure of fluid pressure at all locations, including along interfaces. The associated average pressure for the intersection of the boundary of the phase with the exterior of the domain is

$$p_\alpha^\Gamma = \left\langle p_\alpha \right\rangle_{\Gamma_{\alpha e}, \Gamma_{\alpha e}} \; , \tag{6}$$

for $\alpha \in \mathcal{J}_f$.

The capillary pressure of the two-fluid-phase system depends on the curvature of the interface between the fluids. The curvature of the boundary of phase $\beta$ is defined at the microscale as

$$J_\beta = \nabla' \cdot \mathbf{n}_\beta \; , \tag{7}$$

where $\nabla' = (\mathbf{I} - \mathbf{n}_\beta \mathbf{n}_\beta) \cdot \nabla$ is the microscale divergence operator restricted to a surface, $\mathbf{I}$ is the identity tensor, and $\mathbf{n}_\beta$ is the outward normal vector from the $\beta$ phase. Since the internal boundary is an interface, the curvature of a phase boundary is also the curvature of the interface between phases for locations within the domain. At the microscale, the capillary pressure is defined at the interface between fluid phases as

$$p_{wn} = -\gamma_{wn} J_w \; , \tag{8}$$

where $\gamma_{wn}$ is the interfacial tension of the $wn$ interface. Laplace's law is a microscale balance of forces acting on an interface that relates the capillary pressure to the difference between the microscale phase pressures evaluated at the interface with

$$p_n - p_w = -\gamma_{wn} J_w \; . \tag{9}$$

It is important to understand that Laplace's law applies at points on the $wn$ interface only at equilibrium; the definition of capillary pressure given by Eq. 8 applies even when the system is not at equilibrium. Additionally, if the mass per area of the interface is non-zero, Laplace's law must be modified to account for gravitational effects (Gray and Miller, 2014). Care must be taken when extending this relationship to the macroscale, as is shown below.

Since the capillary pressure is defined for the interface between the two fluids, $\Omega_{wn}$, we consider an average of the microscale curvature based on this entity

$$J_w^{wn} = \left\langle J_w \right\rangle_{\Omega_{wn}, \Omega_{wn}} = -\left\langle J_n \right\rangle_{\Omega_{wn}, \Omega_{wn}} \; . \tag{10}$$

Similarly, the macroscale capillary pressure is

$$p^{wn} = -\left\langle \gamma_{wn} J_w \right\rangle_{\Omega_{wn},\Omega_{wn}} . \tag{11}$$

The case of a constant interfacial tension at the microscale allows for

$$p^{wn} = -\gamma^{wn} J_w^{wn} . \tag{12}$$

In the context of Eq. 9 a third pressure of interest for two-fluid-phase systems is the interface-averaged pressure

$$p_\alpha^{wn} = \left\langle p_\alpha \right\rangle_{\Omega_{wn},\Omega_{wn}} , \tag{13}$$

for $\alpha \in \mathcal{J}_f$. A macroscale version of Laplace's law can then be written as

$$p_n^{wn} - p_w^{wn} = -\gamma^{wn} J_w^{wn}. \tag{14}$$

At equilibrium, Laplace's microscale law will hold everywhere on $\Omega_{wn}$. This implies that Eq. 14 must also be satisfied at
equilibrium for the case of a constant interfacial tension. However, measurements of $p_w^{wn}$ and $p_n^{wn}$ must be performed at
the interface $\Omega_{wn}$. This is not practical, and perhaps not even useful since neither quantity appears in macroscale models.
At the macroscale, it is most convenient to work in terms of averaged phase pressures $p^w$ and $p^n$. Because $p^\alpha$ and $p_\alpha^{wn}$ are
not equivalent, the way in which Eq. 14 can be used is in question. In this work, we explore this dilemma, giving special
consideration to the connectivity of the wetting phase.

In previously published work, we have considered the impact of non-wetting phase connectivity in detail (McClure et al.,
2016b). The connectivity-based analysis presented in that work can be used to re-cast Eq. 14 in terms of the connected wetting
phase regions. These regions are identified by sub-dividing $\Omega_w$ into $N_w$ sub-regions that do not intersect. The sub-regions
cannot touch each other, meaning that $\overline{\Omega}_{w_i} \cap \overline{\Omega}_{w_j} = \emptyset$ for all $i \neq j$ with $i,j \in \{1,2,\ldots N_w\}$ where the overbar on $\Omega$ denotes a
closed domain that includes explicitly the boundary. Interfacial sub-regions are formed from the intersection $\Omega_{w_i n} = \Omega_{wn} \cap$
$\overline{\Omega}_{w_i}$. When the non-wetting phase is fully connected, an approximate version of Laplace's law can be derived as

$$p^n - p^{w_i} = -\gamma^{wn} J_w^{w_i n} , \tag{15}$$

for $i \in \{1,2,\ldots,N_w\}$. This expression relates the average phase pressures within each region of wetting phase to the curvature
of the adjoining interface. The average phase pressures are defined as

$$p^{w_i} = \left\langle p_w \right\rangle_{\Omega_{w_i},\Omega_{w_i}} , \tag{16}$$

and the average curvature as

$$J_w^{w_i n} = \left\langle J_w \right\rangle_{\Omega_{w_i n},\Omega_{w_i n}} . \tag{17}$$

The quantities $p^{w_i}$ and $J_w^{w_i n}$ are averaged quantities, but they are not macroscale quantities. The macroscale pressure of the
wetting phase can be determined as

$$p^w = \frac{1}{\epsilon^{\overline{\overline{w}}}} \sum_{i=1}^{N_w} \epsilon^{\overline{\overline{w_i}}} p^{w_i} , \tag{18}$$

and the macroscale capillary pressure is

$$p^{wn} = -\frac{\gamma^{wn}}{\overline{\overline{\epsilon^{wn}}}} \sum_{i=1}^{N_w} \epsilon^{\overline{\overline{w_i n}}} J_w^{w_i n} \ . \tag{19}$$

For the case where multiple disconnected sub-regions are present for either phase, the relationship between $p^n - p^w$ and $p^{wn}$ is therefore quite complex from a geometric standpoint. Associated challenges for the measurement of phase pressures impact our understanding of the system behavior at the macroscale, hindering our ability to develop effective models.

The definitions of pressures provided demonstrate that several different pressures are of interest for two-fluid systems. In general these pressures will not be equivalent. Thus care is needed in analyzing the system state and in proposing relations among pressures. Typically only the pressure defined by Eq. 6 is measured in traditional laboratory experiments, and this is often true even with state-of-the-science experiments that include high-resolution imaging. On the other hand, computational approaches provide a means to compute all of the defined pressures, yielding a basis to deduce a more complete understanding of the macroscale behavior of the system than would be accessible using approaches that are only able to control and observe fluid pressures on the boundaries of the domain. Further, the formulation detailed above applies for dynamic conditions as well as equilibrium or steady state conditions except where specifically noted. For dynamic conditions, the averaged quantities are computed at some instant in time.

## 5   Materials and Methods

### 5.1   Experimental Design

An experimental approach was sought to investigate the distribution of capillary pressure in a porous medium system. To meet the objectives of this work, we needed directly to observe capillary pressure at high resolution, which requires computation of the average curvature of the fluid-fluid interface as a function of the averaging region. Because we wished to observe systems at true equilibrium and knew from recent experience that extended periods of time are necessary to obtain such a state (Gray et al., 2015), we elected to rely upon a microfluidic approach for which we could verify true equilibrium states were achieved. Microfluidic devices are physically small but can be made sufficiently large to satisfy the conditions for being a valid macroscale REV. This is so because the systems are well above the microscale continuum limit and then only need to satisfy the conditions for the size being a representative sampling of the pore morphology and topology of the media. The size needed for an REV has been investigated previously for two-fluid-phase flow. Typically in three-dimensions a few thousand spheres is needed to produce essentially invariant information for quantities such as saturations, interfacial areas, and capillary pressure. This translates to slightly over 10 mean grain diameters in each dimension. Microfluidic cells can be fashioned to meet this requirement. Even though hydrologic problems motivate this work, the fundamental nature of the capillary pressure state function can be investigated with any pair of immiscible fluids. Minimizing the mutual solubilities of each fluid in the companion fluid is an important design characteristic that can simplify the experimental work without loss of generality. Thus

physically small microfluidic systems that did not include water were used in this work, which might on the surface appear to be far removed from the motivating hydrologic systems of concern.

Experiments involving two-fluid flow through porous media are typically conducted using a setup similar to the one shown in Fig. 1. A porous material, in this case a two-dimensional micromodel cell, is connected to two fluid reservoirs at opposite ends of the sample. The two fluids are referred to as wetting ($w$) and non-wetting ($n$) based on the relative affinity of the fluids toward the solid phase ($s$, the black region of Fig. 1). The two-dimensional micromodel was fabricated using photolithography techniques. The 500 $\mu$m $\times$ 525 $\mu$m $\times$ 4.4 $\mu$m porous medium cell of the micromodel contained a distribution of cylinders, with a porosity of 0.54. The short dimension of the cell was oriented in the vertical dimension such that flow was essentially horizontal. The boundary reservoirs were used to inject fluid into the sample, resulting in the displacement of one fluid by the other. As depicted in Figure 1, one inlet of the cell was connected to a wetting-fluid-phase (decane) reservoir and the other to a non-wetting-fluid-phase (nitrogen gas) reservoir, with the other four boundaries being solid. A displacement experiment was performed in the micromodel depicted in Fig. 1 using the experimental methods detailed in Dye et al. (2015). This approach provides observations of equilibrium configurations of the two-fluid-phase system. The displacement experiment began by fully saturating the porous medium cell with decane through the inlet reservoir located at one end of the cell. Primary drainage was then carried out by incrementally increasing the pressure of the nitrogen reservoir, located on the opposite end of the cell. After each pressure step, the system was allowed to equilibrate. The final equilibrium state for a given pressure boundary condition was determined based on an invariance of the average mean curvature of the $wn$ interface, $J_w^{wn}$, as determined from image analysis. After the system reached an equilibrium state, the pressure in each reservoir, measured with pressure transducers, and an image of the cell were recorded before another incremental change in pressure step was applied. The drainage process was terminated prior to nitrogen breakthrough into the decane reservoir.

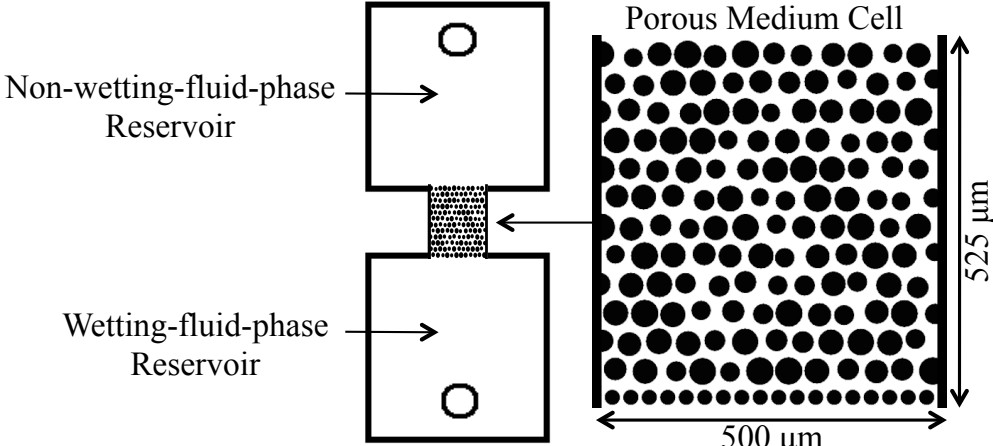

**Figure 1.** A depiction of the two-dimensional micromodel that was used in the displacement experiment. The solid phase consists of pore-space free solid cylinders of varying radii distributed in the horizontal plane represented by black and the regions accessible to fluid flow by white within the porous medium cell.

The solid geometry used in our microfluidic experiments was designed to allow for high capillary pressure at the end of primary drainage. At the wetting-fluid-phase reservoir, a layer of evenly spaced homogeneous cylinders was placed such that the gap between cylinders was uniformly small. This allowed for a large pressure difference between the fluid reservoirs, since the non-wetting fluid phase did not penetrate the wetting-fluid-phase reservoir over a wide range of pressure differences.

## 5.2 Computational Approach

The experimental microfluidics setup described in the previous section provides a way to perform traditional two-fluid-flow experiments and observe the internal dynamics of interface kinematics and equilibrium distributions. Microscale phase configurations can be observed directly, and averaged geometric measures can be obtained from this data. While boundary pressure values are known, the experiment does not provide a way to measure the microscale pressure field. Accurate computer simulation of the experiment can provide this information and can also be used to generate additional fluid configurations that may not be accessible experimentally. In particular, configurations below the so-called irreducible wetting phase saturation will be considered. The common identification of a saturation as "irreducible" is a misnomer because wetting phase saturations beneath this value can be achieved through, for example, evaporation or by initializing a saturation below this value in an experimental setup. In this work, simulation is applied in two contexts: (1) to simulate the microscale pressure field based on experimentally-observed fluid configurations; and (2) to simulate two-fluid equilibrium configurations based on random initial conditions. Success with the first set of simulations in matching the experiments provides confidence that the results of the second set of computations represent physically reasonable configurations. Here we summarize each of the approaches.

Simulations are performed using a "color" lattice Boltzmann method (LBM). Our implementation has been described in detail in the literature (see McClure et al., 2014a, b). The approach relies on a multi-relaxation time (MRT) scheme to model the momentum transport. In the limit of low Mach number, the implementation recovers the Navier-Stokes equations with additional contributions to the stress tensor in the vicinity of the interfaces. The interfacial stresses between fluids result from capillary forces, which play a dominant role in many two-fluid porous medium systems. The formulation relies on separate lattice Boltzmann equations (LBEs) to recover the mass transport for each fluid. This decouples the density from the pressure to allow for the simulation of incompressible fluids. Our implementation has been applied to simulate two-fluid-phase flows in a variety of porous medium geometries, recovering the correct scaling for common curve dynamics (McClure et al., 2016a), and it has also been used to closely predict experimental fluid configurations (Dye et al., 2015; Gray et al., 2015). The effect of gravity was ignored in the simulation of the experimental systems due to the very small length scale in the vertical dimension.

The implementation allows us to initialize fluid configurations directly from experimental images. Segmented images are generated from grey-scale camera data. These images were used to specify the initial position of the phases in the simulations with high resolution. The micromodel cell was computationally resolved within a domain that is $20 \times 500 \times \times 500$. The lattice spacing for the simulation was $\delta x = 1 \ \mu$m. Note that the depth of the micromodel was resolved in the simulation. The physical depth of the simulation cell ($20 \ \mu$m) was larger than the depth of the micromodel cell ($4.4 \ \mu$m). This was done so that the curvature in the depth of the cell could be resolved accurately. Due to geometric constraints, the curvature associated with the

micromodel depth cannot vary. The curvature of the interface between the two fluids can be written as

$$J_w = -\left( \frac{1}{R_1} + \frac{1}{R_2} \right), \qquad (20)$$

where $R_1$ is the radius of curvature in the horizontal plane and $R_2$ is associated with the micomodel depth. Only $R_1$ can vary independently. In the simulation, the fixed value of $R_2$ was 10 $\mu$m. In the experiment, the fixed value of $R_2$ was 2.2 $\mu$m. With $R_2$ known in both cases, the simulated curvatures were mapped to the experimental system. In the experimental system, pressure transducers were used to measure the phase pressures in the boundary reservoirs. These measurements were used to inform pressure boundary conditions within the simulation. Since boundary conditions were enforced explicitly within the simulation, the boundary pressures match the experimentally measured values exactly. The fluid configurations can vary independently based on these conditions. Simulations were performed until the interfacial curvature stabilized, since prior work has demonstrated the important fact that the curvature equilibrates more slowly than other macroscale quantities, such as fluid saturation Gray et al. (2015).

A set of simulations was also performed based on random initial conditions. The approach used to generate random fluid configurations and associated equilibrium states is described in detail by McClure et al. (2016b). The solid configuration for the flow cell was identical for both sets of simulations. Blocks of fluid were inserted into the system at random until a desired fluid saturation was obtained. This allowed for the generation of fluid configurations at wetting phase saturations that were below the experimentally-determined, so-called irreducible wetting-phase saturation. Periodic boundary conditions were then enforced, and the simulation was performed to produce an equilibrium configuration as determined by the average curvature of the interface between fluids. Based on the final fluid configurations, connectivity-based analysis was performed to infer macroscale capillary pressure, saturation, and interfacial area for a dense set of equilibrium states.

## 5.3   Results and Discussion

Phase connectivity presents a critical challenge for the theory and simulation of two-fluid-phase flow. When all or part of a phase forms a fully-connected pathway through a porous medium, flow can occur without the movement of interfaces. However, the case where phase sub-regions are not connected is a source of history-dependent behavior in traditional models. Traditional models make use of the capillary pressure proposed as a function of the fluid saturation only, $p^c(s^{\overline{\overline{w}}})$. However, this relationship is not unique. Furthermore, key features of the relationship are an artifact of the experimental design. For example, the irreducible wetting phase saturation, $s_I^{\overline{\overline{w}}}$, can play an important role.

To calculate $p^w$ as it is defined from Eq. 5, the microscale pressure field must be known throughout the domain. Simulation provides a means to study how the pressure varies within the system and to obtain averages within all phase sub-regions. Based on Eq. 16, values of $p^{w_i}$, $J_w^{w_i n}$ and $\epsilon^{\overline{\overline{w_i}}}$ can be determined for each connected region of the wetting phase $\Omega_{w_i}$ for $i \in \{1, 2, \ldots, N_w\}$. Two sets of simulations were performed, including (1) a set of 24 configurations initialized directly from experimentally-observed configurations along primary drainage; and (2) a set of 48 configurations with random initial conditions as discussed in Section 5.2. The equilibrium fluid arrangements were analyzed to determine the true capillary pressure,

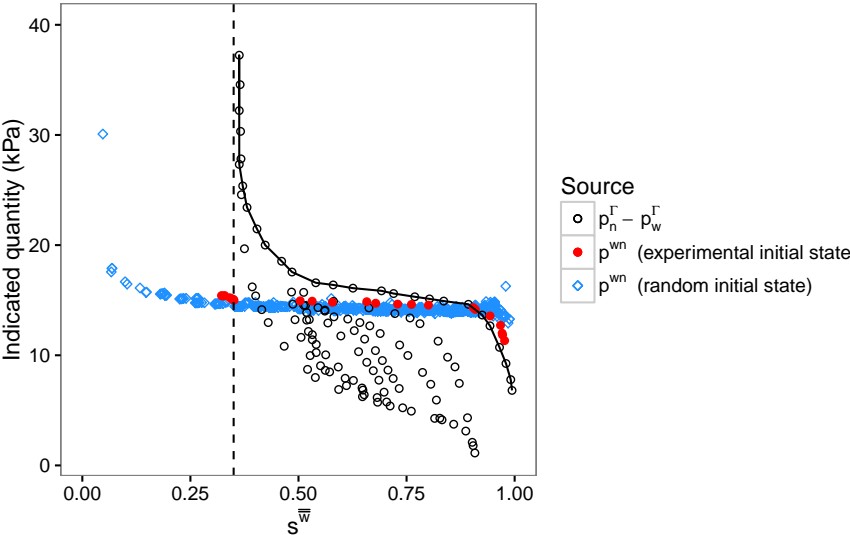

**Figure 2.** Comparison between the experimentally measured boundary pressure difference $p_n^\Gamma - p_w^\Gamma$ and the capillary pressure $p^{wn}$ for the micromodel geometry. The solid line represents the boundary pressure along primary drainage.

$p^{wn}$, by analyzing the curvature of the fluid-fluid interface, fluid saturation, $s^{\overline{\overline{w}}}$, and specific interfacial area, $\epsilon^{\overline{\overline{wn}}}$. The data was aggregated to produce a dense set of equilibrium configurations.

Pressure transducers located in each of the two fluid reservoirs were used to measure experimental boundary pressures for each fluid. The resulting values of $p_n^\Gamma - p_w^\Gamma$ are plotted in Fig. 2. Average capillary pressure values calculated from the
5   simulations are presented along with this experimental data. The solid line represents the boundary pressure difference during primary drainage. The boundary pressures for simulations initialized from experimental data matched the experimentally measured values of $p_n^\Gamma - p_w^\Gamma$ exactly. Boundary measurements taken during simulation are also presented for imbibition and scanning curve sequences. The values of $p_n^\Gamma - p_w^\Gamma$ plotted in Fig. 2 represent a comprehensive set of experimental measurements that would typically be identified as capillary pressure values. This provides a basis for comparison with measurements of the
10  true capillary pressure based on the configuration of the interfaces. In general, agreement between $p_n^\Gamma - p_w^\Gamma$ and $p^{wn}$ should not be expected. Only when both the $w$ and $n$ fluids are fully connected and when the system is at equilibrium will the boundary pressure difference balance the internal average capillary pressure. The difference between the boundary measurement and the internal average capillary pressure due to the phases being disconnected is evident by comparing the experimental data from primary drainage and the simulation points initialized from the associated fluid configurations. Pressure boundary conditions
15  for the simulations were set to match the measured values of $p_n^\Gamma$ and $p_w^\Gamma$. As $s^{\overline{\overline{w}}}$ decreases, there is an increasing gap between $p_n^\Gamma - p_w^\Gamma$ and the average capillary pressure $p^{wn}$. This gap is attributed to the formation of disconnected wetting phase regions during drainage, an effect that is most significant as the so-called irreducible wetting phase saturation is approached.

In the experimental system, an irreducible wetting phase saturation was clearly observed as $s_I^{\overline{\overline{w}}} = 0.35$. This value is marked with a vertical dashed line in Fig. 2. This irreducible wetting phase saturation corresponds to the lowest experimentally accessible wetting phase saturation, since fluid configurations with $s^{\overline{\overline{w}}} < s_I^{\overline{\overline{w}}}$ cannot be obtained from the experimental setup and operating conditions. The underlying reason for this is related to the connectivity of the wetting phase. This can be understood from Fig. 3, which shows the phase configuration observed experimentally at the end of primary drainage. Within a connected region of wetting phase, the microscale pressure, $p_w$, will tend to be nearly constant. However, the wetting phase pressure can vary from one region to another. The connected components of the wetting phase are shown in Fig. 3 (b). At equilibrium, the measured difference in boundary pressures $p_n^\Gamma - p_w^\Gamma$ must balance with the capillary pressure of the interface sub-region between the two phase components. Note that the non-wetting phase is fully connected in Fig. 3 (a). The implication is that $p_n^\Gamma = p^n$ at equilibrium. However, $p_w^\Gamma$ only reflects the pressure of the wetting phase reservoir. The sub-regions of the wetting phase that remain after primary drainage are plotted in color in Fig. 3 (b). The part of $\Omega_w$ that is connected to the wetting phase reservoir is shown in light green in Fig. 3 (b). When the irreducible wetting phase saturation is reached the portion of $\Omega_w$ that connects to the reservoir no longer fills any of the porespace within the micromodel. The irreducible wetting-phase saturation is associated with the trapped wetting phase regions only. Changing the pressure difference between the fluid reservoirs to increase $p_n^\Gamma - p_w^\Gamma$ does not change the capillary pressure in these regions. This leads to arbitrarily high measurements, claimed to be "capillary pressure" measurements, which are actually a difference in reservoir pressures rather than a measure of interface curvature. This also misconstrues the reduction in wetting phase saturation that occurs. The true average capillary pressure, as defined in Eq. 12, is much lower. Furthermore, the wetting-phase saturation can be further reduced as a consequence of other processes, such as evaporation. It is irreducible only within the context of the experimental design.

In light of this result, it is useful to consider alternative means to generate two-fluid configurations in porous media. For example, suppose a fluid configuration were encountered with $s^{\overline{\overline{w}}} = 0.2$, a value lower than the irreducible saturation. How can we determine the macroscale capillary pressure? From a traditional macroscale parameterization approach, the experimentally proposed relation $p^{wn}(s^{\overline{\overline{w}}})$ is of absolutely no use, since capillary pressure is undefined for $s^{\overline{\overline{w}}} < s_I^{\overline{\overline{w}}}$. From the microscale perspective, it is clearly possible to produce fluid configurations for which $s^{\overline{\overline{w}}} < s_I^{\overline{\overline{w}}}$ (for any system), and to measure the associated capillary pressure based on Eq. 12. For randomly initialized phase configurations, many such systems are produced. Simulations performed based on these initial geometries lead to equilibrium capillary pressure measurements shown in Fig. 2. While the classic "J curve" shape is still present, the experimentally-determined value $s_I^{\overline{\overline{w}}}$ offers no guidance regarding this form.

Comparing capillary pressures measured from random initial conditions with those measured from experimental initial conditions provides additional insight. First, the true capillary pressure measurements based on Eq. 8 are remarkably consistent, particularly when considering the values of $p^{wn}$ obtained as $s^{\overline{\overline{w}}} \to s_I^{\overline{\overline{w}}}$. Compared to randomly initialized data, configurations from primary drainage have a higher average capillary pressure. This is expected, since along primary drainage $p^{wn}$ is determined by the pore-throat sizes. These represent the highest capillary pressures that are typically observed. We note that primary drainage does not specify the maximum possible capillary pressure, since bubbles of non-wetting phase may form that have a smaller radius of curvature than the minimum throat width.

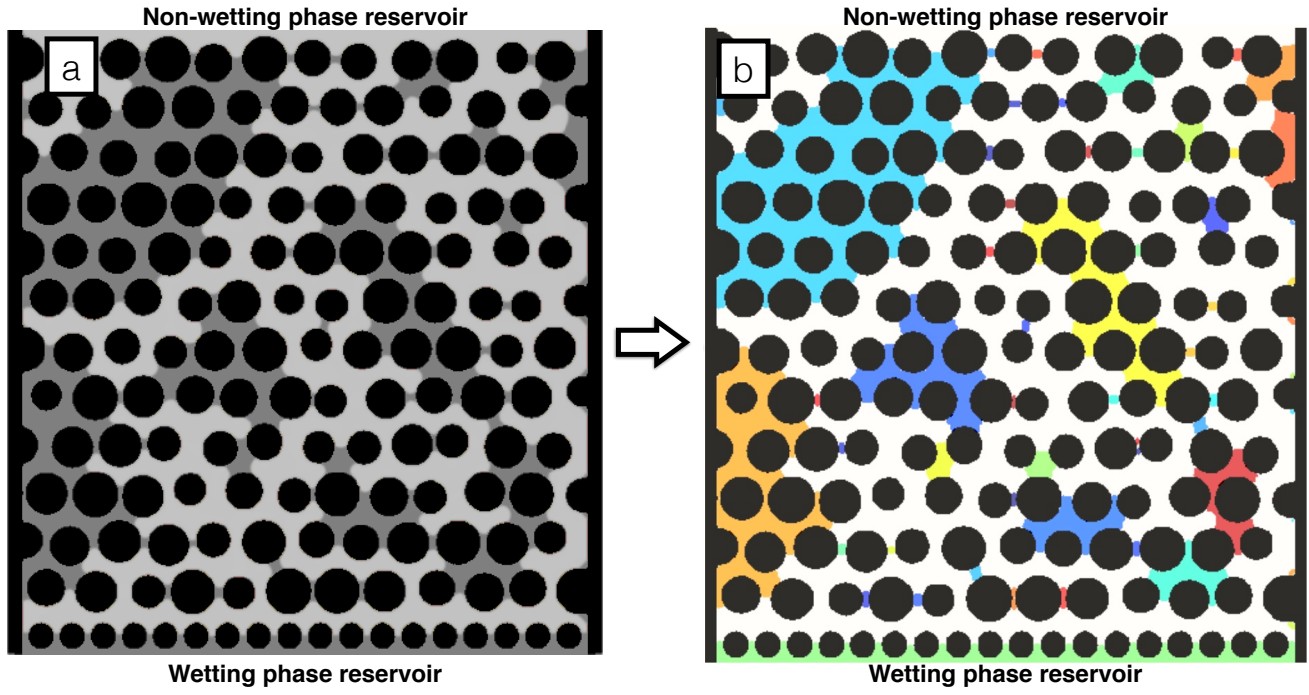

**Figure 3.** Phase connectivity has a direct impact on the meaning of the macroscale experimental measurements: (a) experimentally observed phase configuration corresponding to irreducible wetting phase saturation; and (b) connected components analysis shows all wetting phase that remains in the system is disconnected from the wetting phase reservoir. The black denotes the solid phase, the gray and various colors denote the wetting phase, and the white denotes the non-wetting phase.

Since the boundary pressure difference $p_n^\Gamma - p_w^\Gamma$ cannot be substituted for the capillary pressure, a key question is how this impacts capillary pressure hysteresis. When $p_n^\Gamma - p_w^\Gamma$ is used to erroneously infer the capillary pressure, the relationship between capillary pressure and saturation appears as the black circles in Fig. 2. When the true capillary pressure is used to plot the same data the shape of the relationship between capillary pressure and saturation is distinctly different. Capillary pressures are

5   obtained at all fluid saturations, and no irreducible wetting-phase saturation is observed. Due to the fact that the true capillary pressure includes the effects of disconnected phase regions, moderate capillary pressures are observed. This is because the extrema for the boundary pressure measurements are not constrained by the internal geometry. We note that the relationship $p^{wn}(s^{\overline{\overline{w}}})$ remains non-unique, since capillary pressure is not a one-to-one relationship with wetting-phase saturation. The higher-dimensional form $p^{wn}(s^{\overline{\overline{w}}}, \epsilon^{\overline{wn}})$ is therefore considered in Fig. 4. Using a generalized additive model (GAM) (Wood,

10   2008), a best-fit surface was generated to approximate the simulated data, incorporating data points derived from both random and experimentally-observed initial conditions. The black lines in Fig. 4 show the iso-contours of the capillary pressure surface. It is clear that primary drainage leads to states with lower interfacial area as compared to randomly initialized configurations.

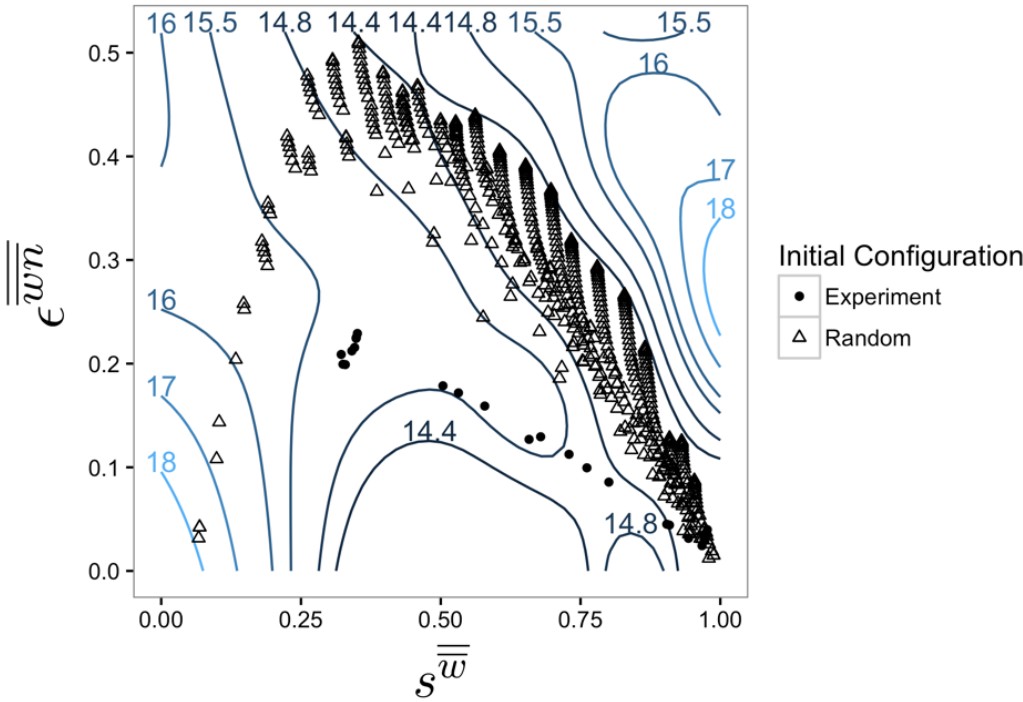

**Figure 4.** Contour plot showing the relationship $p^{wn}(s^{\overline{w}}, \epsilon^{\overline{\overline{wn}}})$, with contours showing the capillary pressure $p^{wn}$ (kPa). Data points used to construct the surface are also shown, including randomly initialized fluid configurations and experimentally initialized configurations from primary drainage.

Both sets of points lie along a consistent surface. The extent to which the relationships $p^{wn}(s^{\overline{\overline{w}}})$ and $p^{wn}(s^{\overline{\overline{w}}}, \epsilon^{\overline{\overline{wn}}})$ describe the data points measured from microscale configurations is quantitatively assessed by evaluating the residuals for the GAM approximation. The residuals are shown in Fig. 5. The traditionally used relationship $p^{wn}(s^{\overline{\overline{w}}})$ is able to explain only 60.6% of the variance in the data. When the effect of interfacial area is included, $p^{wn}(s^{\overline{\overline{w}}}, \epsilon^{\overline{\overline{wn}}})$, 77.1% of the variance is explained. Based on previous work for three-dimensional porous media, it is anticipated that higher fidelity approximations can be produced by including the effects of other topological invariants, such as the average Gaussian curvature or Euler characteristic (McClure et al., 2016b).

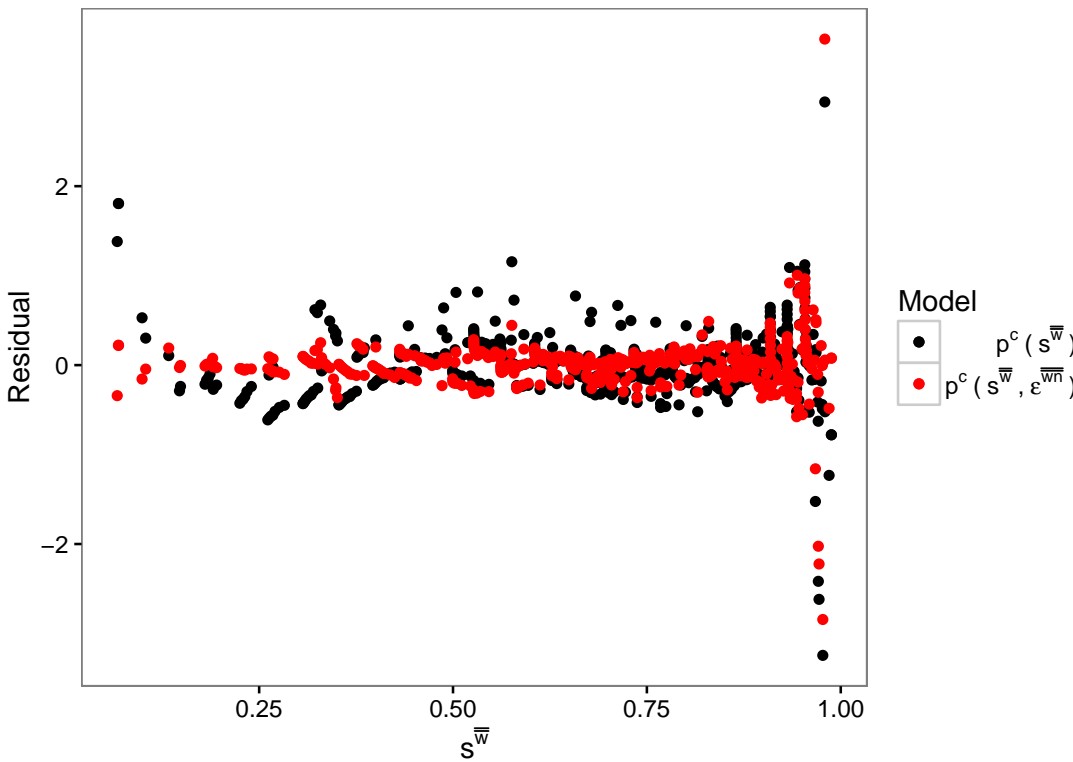

**Figure 5.** Comparison of the residual errors for the GAM fits that approximate $p^{wn}(s^{\overline{\overline{w}}})$ and $p^{wn}(s^{\overline{\overline{w}}}, \epsilon^{\overline{\overline{wn}}})$.

## 6 Conclusions

In this work, we show that the ability to quantitatively analyze the internal structure of two-fluid porous medium systems has a profound impact on macroscale understanding. We considered the behavior of the capillary pressure based on traditional laboratory boundary measurements and compare this to the true average capillary pressure, a state function, determined by directly averaging the curvature of the interface between fluids. We demonstrate that the difference between the phase pressures as measured from the boundary cannot be used to deduce the capillary pressure of the system. In particular, the high capillary pressure measured for irreducible wetting phase saturation is an artifact of the experimental design. Four important conclusions result.

First, the true capillary pressure measured at traditionally identified irreducible wetting-phase saturation is significantly lower than predicted from boundary pressure measurements. This can be understood based on the underlying phase connectivity. At irreducible wetting-phase saturation, the wetting-phase reservoir pressure no longer reflects the internal pressure of the system since the reservoir does not connect to the remaining wetting phase inside the system.

Second, randomly generated fluid configurations provide a way to access states where the wetting-phase saturation is below the irreducible wetting phase saturation. By carrying out direct averaging based on these states, the capillary pressure state

function can be computed over the full range of possible saturation values, including configurations that are inaccessible from traditional experiments. We note that modified experimental designs could be used to accomplish the same studies.

Third, we show that the equilibrium relationship among capillary pressure, fluid saturation and interfacial area is consistent between randomly initialized configurations used only in computation and experimentally initialized configurations. Combining the two data sets, generalized additive models were used to approximate the surface relating $p^c$, $s^{\overline{\overline{w}}}$, and $\epsilon^{\overline{\overline{wn}}}$. At fixed saturation, states evolved from primary drainage have higher capillary pressure and lower interfacial area than equilibrium states that evolve from randomly generated states. Our results are particularly significant for systems where low wetting-phase saturations are important, such as evaporation in the vadose zone.

*Author contributions.* All authors participated in the writing of this manuscript. WGG and CTM contributed to the introduction, background, and theory, ALD contributed to the microfluidics, and JEM contributed to lattice Boltzmann modeling. All authors contributed to the discussion and conclusions from this work.

*Acknowledgements.* This work was supported by Army Research Office grant W911NF-14-1-02877, Department of Energy grant DE-SC0002163, and National Science Foundation grant 1619767. An award of computer time was provided by the Department of Energy INCITE program. This research also used resources of the Oak Ridge Leadership Computing Facility, which is a DOE Office of Science User Facility supported under Contract DE-AC05-00OR22725.

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
