# Peer review of "On the Consistency of Scale Among Experiments, Theory, and Simulation"

_Hydrology and Earth System Sciences, 2016_

## Referee Comment (RC1) · Anonymous Referee #1 · 29 Sep 2016

The paper critiques current models and makes a case for developing models that are consistent across scales based on thermodynamic principles. The nature of the processes these models tackle is kept vague, but some hints suggest that models for subsurface water flow (soil water and groundwater) are the prime target. A theoretical treatment of the Laplace Law is developed to develop equations for microscale capillary pressures, which seems to refer to pressure jumps across fluid-fluid interfaces in single pores. These expressions are more general that the Laplace Law because they apply to equilibrium and non-equilibrium cases. Expressions for average intrinsic phase pressures are also presented.

An experiment is described in which a non-wetting gas phase (nitrogen gas) permeates a 0.5 by 0.5 mm two-dimensional porous medium saturated by a wetting fluid phase (decane). This process and similar ones with different initial and boundary conditions

are also modeled numerically.

Both the simulated and observed data are used to obtain the 3D equivalent of the decane retention function in which the degree of saturation is a function of both the average fluid pressure and the specific interface area.

Major comments

For a paper on scales I could not help noticing that the time scale is mentioned only once and that there is no clear definition of the spatial scales of interest (microscale and macroscale). No connection is established between these scales and the scale of the representative elementary volume.

The paper uses a few straw man arguments. It is claimed that in experiments, pressures are only measured (or set) at the boundary of the system of interest. With the increased use of microtensiometers this is no longer necessarily the case. In my experience (and with some support in the literature), the microtensiometers tend to confirm that the known pressure at a boundary can be used to calculate the pressure anywhere in the system as long as contact is good and equilibrium has been achieved. The reliance on boundary pressures is not as risky as the authors appear to believe. In the terminology of the analysis of the paper this implies that phase continuity in real-world porous media is often sufficient for the observed pressures to be valid.

The authors state that average phase pressures are convenient to work with. I have never read anything in support of this argument. There are no sensors to measure average pressures, so we cannot calibrate models on them, and I have not come across any work that used average pressures in lieu of local pressures and pressure gradients.

I have the impression that the analysis is valid for zero-gravity conditions. This is never stated explicitly, but three elements of the paper suggest it:

- the casual averaging of pressures without acknowledging the immense effect of the geometry of real-world fluid bodies on the average pressure when gravity is non-zero

[Figure]

- the implicit notion that fluid interfaces and common curves have a non-zero thickness and therefore mass, without the effect of this mass being discussed or even mentioned.

- the extremely small size of the porous medium used in the experiment that indeed makes the effect of gravity negligible. In a paper in which the introduction discusses the importance of consistency of scales for scale ranges that are many orders of magnitude larger and already in the abstract calls for models that are based on rigorous multiscale principles this severely limits the relevance of the paper.

The lack or relevance is further reduced by the experimental scale: 0.25 square millimeter is in the sub-Darcian scale for most soils and geologic materials. To call this scale the macroscale seems to betray a fundamental lack of understanding of the concepts of the continuum approach and the representative elementary volume that form the basis that most currently used models are founded on.

Section 4 'Approach' has a non-informative title. It can easily be split in a 'Theory' section (modify the title as desired) and a 'Materials and Methods' section, thereby making the paper conform to the established structure of scientific papers. The Results and Discussion section is already there.

Section 4 starts with a treatment of the Laplace Law. One of the authors published an extensive treatment of this law (Hassanizadeh and Gray, 1993, not quoted in the paper). I would like to see included in this paper an explanation of the added value of the current discussion in view of this earlier work, and how this treatment relates to that in the earlier work. There are marked distinctions in notation between the earlier and the current paper which made it hard for me to establish the relation.

The work culminates in a relationship between capillary pressure, degree of saturation, and specific interfacial area. As long as the latter cannot be measured on 3D samples, the work has no chance of becoming applicable.

I do not see a path for using this kind of work to arrive at the thermodynamically consistent, scalable models for porous media found in nature, even though the authors claim that goal to be a main motivation for the paper.

Overall assessment

The paper has six objectives that claim to resolve several issues relating to capillary pressure at the micro- and the macroscale and expose limitations of conventional approaches.

The Introduction and its list of objectives raise high expectations about the impact and relevance of this paper for modeling of multiphase flows in soils, aquifers, oil deposits, etc. These expectations are in no way met, either by the theoretical analysis that adds only incrementally to an earlier paper and omits gravity, or by the experiment on 0.25 square mm of an artificial, two-dimensional porous medium with two fluids that have no relevance for hydrology. To make the contrast between this work and real-world hydrology even more glaring, the authors drop the name of Eric Wood, who has worked on continental and global hydrology.

The presentation of the material is messy:

- the Introduction dwells on subjects not at all covered by the paper and fails to inform the reader about the paper's focus and nature of the work.

- the list of objectives is too long, and vastly overstates what the paper actually delivers.

- the paper is not well structured - there is no Materials and Methods section, and the flow of thought is sometimes hard to follow. Some parts are well written, others much less so. A strict adherence to the established format of a scientific paper would help.

- not all variables and symbols are explained, and there are inconsistencies in the notation

- the description of the experiment and the computations (what should be the Materials and Methods section) is incomplete.

Detailed comments are given in the file accompanying this review.

Reference: Hassanizadeh, S.M., and W.G. Gray, Thermodynamic basis of capillary pressure in porous media, Water Resour. Res. 29:3389-3405, 1993.

Please also note the supplement to this comment:
http://www.hydrol-earth-syst-sci-discuss.net/hess-2016-451/hess-2016-451-RC1-supplement.pdf

[Figure]

**Supplement:**

[revised manuscript text omitted]

---

## Referee Comment (RC2) · Anonymous Referee #2 · 10 Oct 2016

in this manuscript, physically based upscaling of two phase fluid flow in a porous medium is considered by presenting definitions of microscopic and (macroscopic) averaged properties, and investigating this system with experiments and simulations. The manuscript provides nice illustrations of how different experimentally determined pressure differences and local values of capillary pressure are. This is done by a blend of experiments and numerical simulations. While I have no problem with the basic message of the manuscript, the presentation is not as may be expected. Quite some space is reserved for the objectives, a literature overview in the background section, and the presentation of eqs. (1)-(19), which are basically definitions. What remains underexposed, though, is a clear identification of what is new. Certainly, averaging is not, and neither is it for two fluid systems in porous material. Therefore, I propose that this is explicitly mentioned on these sections 2-4.1, as I am not convinced that these

sections should be maintained in this manuscript. The aspect of connectivity is given some emphasis (e.g. p.8) and reference is made to McClure et al. Again, I propose that it is clearly identified whether and what is new in this work, as the current text is not clarifying this. Later on, again the experimental and simulation parts appear to be based on work of McClure et al. and it is apparent that this work may duplicate that earlier work. Though the present manuscript is illustrative, I would consider it not fit for publication, if in essence the material is a duplication of earlier work. One of the issues that is quite central to this manuscript is that equilibrium is achieved. Considering the small size of the apparatus, I wonder how this is checked. On several other statements I also wonder what their justification is. Presumably, this is indicated in the cited references, but as a stand-alone manuscript, important statements need to be justified here. specific comments: 1. I wonder about some of the English (is the term microfluidic well used; abstract; these instances on p.4 line 11). The abstract contains quite some text, which I would rate as context, that is not necessary for an abstract and must be deleted: lines 1-8 or even 1-11. In addition, the reduction of water content to below the irreducible saturation is mentioned: As the authors make a call for rigorous definitions, I think this contradiction in the text is inappropriate. Of course, in a special issue focused on Eric Wood, there is a temptation to give some thoughts on his career. However, in this manuscript, those thoughts look quite artificial and unnatural. I would omit those parts of the text. 2. Averaging (p.4) is older. For instance De Josselin de Jong (around 1955) 3. p.5 line 2: I would add: does not ONLY depend... 4. I do understand neither the notation nor the meaning of (2) or the term 'extent measure'. Please clarify. 5. page 8: the term averaged phase pressures is used. I think that it is not appropriately, especially for this manuscript, to be vague about 'over what is averaged'. 6. in Fig.1, the black circles represent solid phase particles. Are these in fact porous cylinders as I understand from p.9 line 15? I think this info should be made very explicit, to address whether or not this experiment is true 2D or in fact 3D (with additional complications that will be obvious). One complication that may not be left undiscussed is that of boundary effects (at front and rear plates). In the same context,

I do not understand p.11 line 4-5: why the 'depth' (in Fig.1: vertical, horizontal,...) of the real apparatus and of the model differ. 7. Is the instrument new? I ask because it is not clear whether the experiments, their interpretation and such are new and in what sense (see p.10 line 17-18). 8. you create random initial conditions below irreducible saturation (p.11). Only now, it is indicated clearly what makes it irreducible: because it is not connected to the wetting phase reservoir. I think that this needs to be mentioned earlier. Also, explain why it is relevant: these situations cannot develop in reality (for the experimental set up) as it is a state below irreducible. You mention (p.11) that below irreducible saturation, where sub-regions are unconnected, this leads to history dependence. I would think that the same is true in the random initial conditions simulations. Where you inject your 'blocks', is simulating history.

---

## Author Response (AR1)

Response to Review #1 of
**On the Consistency of Scale Among Experiments, Theory, and Simulation**
J.E. McClure, A.L. Dye, W. G. Gray, and C. T. Miller

hess-2016-451

**1   General**

We respond to the comments from Referee #1 beneath comments made. The authors' response is shown as **AU: red**. The changes made are highlighted as **AU: blue**.

**2   Referee #1**

The paper critiques current models and makes a case for developing models that are consistent across scales based on thermodynamic principles. The nature of the processes these models tackle is kept vague, but some hints suggest that models for subsurface water flow (soil water and groundwater) are the prime target. A theoretical treatment of the Laplace Law is developed to develop equations for microscale capillary pressures, which seems to refer to pressure jumps across fluid-fluid interfaces in single pores. These expressions are more general that the Laplace Law because they apply to equilibrium and non-equilibrium cases. Expressions for average intrinsic phase pressures are also presented.

An experiment is described in which a non-wetting gas phase (nitrogen gas) permeates a 0.5 by 0.5 mm two-dimensional porous medium saturated by a wetting fluid phase (decane). This process and similar ones with different initial and boundary conditions are also modeled numerically. Both the simulated and observed data are used to obtain the 3D equivalent of the decane retention function in which the degree of saturation is a function of both the average fluid pressure and the specific interface area.

**Major comments**

For a paper on scales I could not help noticing that the time scale is mentioned only once and that there is no clear definition of the spatial scales of interest (microscale and macroscale). No connection is established between these scales and the scale of the representative elementary volume.

AU: The time scale does not affect the form of the equations relied upon in this work. In fact, this work is primarily concerned with equilibrium states, how to best explore the potential states that can exist, and how the state function of capillary pressure can be represented. Time scales are, however, mentioned in the introduction, the background, and the results sections. The reviewer is mistaken that the microscale and macroscale are not connected, as all of the macroscale quantities defined and used in this work are defined completely and explicitly in terms of microscale quantities—thus making the connection that the reviewer claims is missing. We could add explicit definitions of the microscale and the macroscale terminology that we use in this manuscript, and incorporate additional discussion on the scale required to obtain a representative elementary volume.

AU: We have added a paragraph at the beginning of the Background section to explain the differences between the microscale and the macroscale.

The paper uses a few straw man arguments. It is claimed that in experiments, pressures are only measured (or set) at the boundary of the system of interest. With the increased use of microtensiometers this is no longer necessarily the case. In my experience (and with some support in the literature), the microtensiometers tend to confirm that the known pressure at a boundary can be used to calculate the pressure anywhere in the system as long as contact is good and equilibrium has been achieved. The reliance on boundary pressures is not as risky as the authors appear to believe. In the terminology of the analysis of the paper this implies that phase continuity in real-world porous media is often sufficient for the observed pressures to be valid.

AU: We agree with the reviewer that microtensiometers provide a means to measure fluid pressures within a domain, and we can note this in a revised manuscript. We also agree with the reviewer that if both fluids are well connected across an experimental cell and at a true equilibrium state, the boundary condition measurements and microtensiometer observations should be in agreement. We disagree with the reviewer that such observations are adequate for characterizing the state of a porous medium system in a general sense, and the results presented in this manuscript clearly support our view. For example, imbibition is well-known to result in disconnected non-wetting phase regions, which will not be connected to the boundaries; the formation of disconnected pendular rings of wetting phase is also well-known. Only if sufficient observations of the pressures of each of the disconnected regions and their morphological characteristics were available would the state of the system

be adequately characterized.

This does not mean that the associated capillary pressures are inaccessible from experiment. On the contrary, the increased use of x-ray micro-computed tomography ($\mu$CT) makes it possible to directly measure the interfacial curvature within 3D experimental systems. This approach has been used for about 20 years and is now used routinely [e.g., 1, 3, 2]. As stressed in the manuscript, the true capillary pressure is the product of the average curvature and the interfacial tension. The average curvature can be determined directly from experimental $\mu$CT.

AU: We have added a paragraph at the beginning of section 4.1 to clarify issues associated with pressure.

The authors state that average phase pressures are convenient to work with. I have never read anything in support of this argument. There are no sensors to measure average pressures, so we cannot calibrate models on them, and I have not come across any work that used average pressures in lieu of local pressures and pressure gradients.

AU: Several measures of pressures are important and come directly out of the TCAT theory. These include volume-averaged pressures, interface-averaged pressures, and pressure averaged over a boundary of a system, such as is the case with conventional pressure-saturation experiments. Common existing measurement methods provide averaged quantities due to the size of the instrument. Mechanistic conservation of momentum models include volume-averaged pressures, so from this perspective such quantities are convenient to deal with. Models are developed based on equations that use average pressures and thus must be calibrated and validated in terms of average pressures. It is precisely the distinctions among the different measures of pressures that are a key aspect of the phenomena explored in this work. For theory, models, and data to be mutually useful, they must have a common usage and understanding of pressure. We can highlight these points in a revision of this work.

AU: These issues have now been addressed in text added at the beginning of section 4.1.

I have the impression that the analysis is valid for zero-gravity conditions. This is never stated explicitly, but three elements of the paper suggest it:

- the casual averaging of pressures without acknowledging the immense effect of the geometry of real-world fluid bodies on the average pressure when gravity is non-zero?

- the implicit notion that fluid interfaces and common curves have a non-zero thickness and therefore mass, without the effect of this mass being discussed or even mentioned.
- the extremely small size of the porous medium used in the experiment that indeed makes the effect of gravity negligible. In a paper in which the introduction discusses the importance of consistency of scales for scale ranges that are many orders of magnitude larger and already in the abstract calls for models that are based on rigorous multiscale principles this severely limits the relevance of the paper.

**AU: The TCAT theory relied upon in this work includes the effects of gravity in large systems and for interfaces that contain mass; references to this theory are provided. The formulation provided is not affected by the importance of gravity, and all equations hold regardless of the importance of gravity. In this work, gravitational effects were considered to be negligible due to the size of the system, which we will be sure is clearly noted in a revised version.**

**AU: We have noted that the simulations of the experimental systems neglected gravitational effects, which follows from the very small length scale in the vertical direction. This revision is noted in section 5.2.**

The lack or relevance is further reduced by the experimental scale: 0.25 square millimeter is in the sub-Darcian scale for most soils and geologic materials. To call this scale the macroscale seems to betray a fundamental lack of understanding of the concepts of the continuum approach and the representative elementary volume that form the basis that most currently used models are founded on.

**AU: The reviewer is mistaken. The actual physical size of a system is not an appropriate measure of whether a system is an REV or not. Karst systems may require 100's of meters for a valid REV, whereas microfluidic systems of the sort relied upon in this work can satisfy the physical and mathematical requirements for an REV at length scales on the order of 500 $\mu m$ or less quite easily. At the microscale, the laws of continuum mechanics apply for a fluid at length scales that are long compared to the mean free path between molecular collisions. For the particular system investigated, the continuum limit would be easily satisfied with a length scale of 1 $\mu m$. A valid macroscale requires a clear separation of length scales with the microscale and the resolution scale needed to characterize the pore morphology and topology. This scale usually translates to systems with a length of at least 10 mean grain diameters on a side. While the systems investigated are physically small, they are close**

to an REV in size. The actual physical size cannot be examined in isolation in reaching conclusions about whether a system is an REV. The systems investigated in this study were sufficiently large to show the occurrence of many regions of disconnected phases, which was sufficient to investigate the state function for capillary pressure. We would add some minor discussion about the size of an REV for porous medium systems and reference these comments to the literature.

AU: We have included a paragraph at the beginning of the experimental section 5.2 that explains REV and fluid issues related to the microfluidic work and supported our claims with references to the literature.

Section 4 "Approach" has a non-informative title. It can easily be split in a "Theory" section (modify the title as desired) and a "Materials and Methods" section, thereby making the paper conform to the established structure of scientific papers. The Results and Discussion section is already there.

AU: This appears to be a matter of style preference. Information in the text and cited references provide sufficient background such that experiments used in this work could be reproduced. We note that the HESS guidelines for manuscript preparation do not explicitly require a "Materials and Methods" section. Our manuscript conforms to the structure established in the HESS guidelines. For a serious reviewer to imply that a paper that does not use his or her preferred section headings violates the "established structure of scientific papers" is astonishing.

AU: While we don't agree with the comments of the reviewer, we nonetheless changed the headings as suggested, because it did not degrade the quality of the paper to do so.

Section 4 starts with a treatment of the Laplace Law. One of the authors published an extensive treatment of this law (Hassanizadeh and Gray, 1993, not quoted in the paper). I would like to see included in this paper an explanation of the added value of the current discussion in view of this earlier work, and how this treatment relates to that in the earlier work. There are marked distinctions in notation between the earlier and the current paper which made it hard for me to establish the relation.

AU: A quick search on Google for "capillary pressure porous media" provides over 1M hits. A similar search in Google Scholar provides almost 0.25M hits. We can hazard a guess, with confidence, that many of these papers have made useful contributions to the study of porous media. It is clear that the authors have not seen fit to cite

much of this wealth of information. Even the authors of this paper have been engaged in a good number of papers that deal with porous media physics and capillary pressure. We have chosen not to cite most of these as well because they are tangential to the mission of the current paper. We can say, with confidence, that our work through the years has demonstrated a development in theory and understanding. We have not been stagnant and insisted on sticking with theories and understandings that have become dated or outmoded. Indeed, reports on developments of new theories, experimental tools, experimental techniques, and simulation algorithms do not necessarily provide reports on or references to older methods that the current work supersedes. For the case at hand, the reviewer seems to admire the 1993 paper, and we appreciate that. This 23 year old paper made a contribution at that time. The discussion of the microscale capillary pressure is informative. Frankly, the discussion of the macrocale capillary pressure has been surpassed by understanding gleaned from careful development and application of the TCAT method. This does not negate the contribution of the older paper; neither would a comparable statement about any of the 0.25M citations dealing with capillary pressure in porous media negate their contributions. We can suggest that the reviewer might benefit from looking at more recent contributions in this area of study. We believe that citing one's own work can be self-serving when that work is dated and not particularly pertinent to the issue or issues under discussion in a newer work. We prefer to include references that best serve the hydrologic community that seeks to understand what we are working on and the nature of our contributions. For this reason, we do not cite the 23 year old paper; neither do we provide an extensive review of developments in understanding of capillary pressure, particularly at the macroscale, over the same period. We have a focused set of objectives we wish to address in this manuscript; we employ theoretical, experimental, and computational approaches for doing so. We cite references that are helpful and/or fundamental to fulfilling the objectives of our paper. We see no technically sound reason to cite the paper the reviewer refers to.

The notation used here is explained carefully, and references are given to other works where this notation is explained in detail and used for a variety of applications. Indeed, even when a work involves a more advanced and precise notation than previously used, authors do not have a responsibility to explain and account for the myriad of notations that are used in the same field or in earlier incarnations of work.

AU: No changes were made in response to this comment.

The work culminates in a relationship between capillary pressure, degree of saturation, and specific interfacial area. As long as the latter cannot be measured on 3D samples, the work has no chance of becoming applicable.

**AU: We disagree with the reviewer, who seems to be unaware of the considerable amount of active research in this area. Specific interfacial areas are indeed routinely measured in 3D samples now. We present simulations in this work where those quantities are evolved and compare virtually identically with experimental observations. Fast imaging methods are now capable of measuring specific interfacial areas dynamically and nondestructively. The state equation for capillary pressure depends upon a sufficient set of measures of the morphology and topology of the pore space, along with fluid and solid properties. There is no question that specific interfacial area is one of these quantities, as has been well established in the literature. We would like to add that the functional dependence we propose is correct. In itself, that is important. In practice, one does not discard a correct theory in favor of an incorrect one simply because quantities in the correct theory may be difficult to measure. In the present case, the theory is correct, and the results of the combined theoretical, experimental, and computational studies in this paper are moving the theory forward to becoming applied and employed.**

**AU: No changes were made in response to this comment.**

I do not see a path for using this kind of work to arrive at the thermodynamically consistent, scalable models for porous media found in nature, even though the authors claim that goal to be a main motivation for the paper.

**AU: The reviewer may not see the path; but it clearly exists. We have cleared most of the brush obstructing it. Many visionary researchers are making progress in obtaining appropriate scalable models for porous media. We can caution that some researchers have claimed to have a model that is "thermodynamically consistent" that, in fact, is not. The work here provides a different and correct direction. In general, we do not think that determinations on whether research should be conducted or presented should be based on the suspicion of one individual (or a few or even many individuals) who claims to have limited vision. The reviewer provides no concrete comments based upon scientific observations but only identifies his/her lack of vision and chooses only to speculate idly.**

**AU: No changes will be made in regard to this speculation.**

**Overall assessment**

The paper has six objectives that claim to resolve several issues relating to capillary pressure at the micro- and the macroscale and expose limitations of conventional approaches.

The Introduction and its list of objectives raise high expectations about the impact and relevance of this paper for modeling of multiphase flows in soils, aquifers, oil deposits, etc. These expectations are in no way met, either by the theoretical analysis that adds only incrementally to an earlier paper and omits gravity, or by the experiment on 0.25 square mm of an artificial, two-dimensional porous medium with two fluids that have no relevance for hydrology. To make the contrast between this work and real-world hydrology even more glaring, the authors drop the name of Eric Wood, who has worked on continental and global hydrology.

**AU: We disagree with the reviewer. On page three we list six objectives, each of which is clearly addressed in the material that follows. Because this manuscript was submitted to be part of a special issue in honor of Professor Wood, it seems appropriate to link this work with the work of Professor Wood. His work and this manuscript deal with issues of scale in hydrologic systems. We believe the treatment and tribute is appropriate. We can add that the theoretical approach that is employed here for small systems can be and has been employed for larger systems, including surface hydrology systems. The overriding common thread is "change of scale." The tools for achieving this are the same, the applications are different.**

The presentation of the material is messy:

- the Introduction dwells on subjects not at all covered by the paper and fails to inform the reader about the paper's focus and nature of the work.
  **AU: The Introduction purposely links issues of interest to a broader community to the issue of scale as important for porous medium systems. The present form seems appropriate given the nature of the special issue.**
- the list of objectives is too long, and vastly overstates what the paper actually delivers.
  **AU: The list of objectives is short and each objective is accomplished in the text that follows. It is not clear what the reviewer finds to be overstated or undelivered.**
- the paper is not well structured - there is no Materials and Methods section, and the flow of thought is sometimes hard to follow. Some parts are well written, others much less so. A strict adherence to the established format of a scientific paper would help.
  **AU: We have adhered to the format established by the guidelines**

for manuscript preparation that are available from HESS online. **AU: We have changed the section titles as suggested in this revision.**

- not all variables and symbols are explained, and there are inconsistencies in the notation

  **AU: This comment is again made without evidence. What inconsistencies? What isn't defined? We have attempted to ensure that each variable is defined. If we have missed any or some, we regret that and would be delighted to address that oversight. We will double check the notation. The reviewer could provide a service by identifying any issues he/she has discovered.**

  **AU: We have double checked and also examined the reviewer's marked manuscript. All variables are defined.**

- the description of the experiment and the computations (what should be the Materials and Methods section) is incomplete.

  **AU: Additional details on the experimental methods could be added, although the methods are standard and have been previously published. We don't believe these additions are necessary, and think they would distract from the thrust of the paper and unnecessarily lengthen it. References to experiments and computations are provided. The reviewer seems to be hung up on some preconceived notion of the organization of a scientific contribution that seems to overwhelm his/her ability to assess the actual contents of the contribution. We prefer not to add more details on the experiments and computations as this would be redundant and would add unnecessary clutter to the literature.**

  **AU: Some additional details have been added to the experimental section to explain issues related to the REV and choice of fluids.**

**AU: The authors will be happy to go through the manuscript and consider changes that might be appropriate in light of the review comments and our own view of the work. We note that this reviewer made a number of detailed and useful comments in an attached document, and we welcome the opportunity to consider incorporating this information in a revision of this manuscript.**

**AU: In response to comments made in the marked manuscript but not specifically detailed in these reviewer comments, we have made the following changes: (1) we have substantially revised the abstract; (2) we added two sentences to make the structure of the introduction more clear; (3) modification of the statement on the role of theory was broadened; (4) the role of statistical approaches has been softened; (5) formatting for references has been cleaned up; (6) additional definition of terms where indicated have been added; (7) many minor suggested edits made; (8) additional details were**

added on the formulation of specific interfacial areas and common curve lengths; (9) the index set of fluid phases was defined; (10) additional comments on the limitations of pressure transducers for measuring the desired state of a system were added; (11) a reference was added where the reviewer was confused about how to include gravity in Laplace's law; (12) a sentence was added that all equations apply to dynamic conditions as well as static conditions, except for those specifically noted to require equilibrium conditions for which the velocity is zero by definition; (13) the symbol $\overline{\Omega}$ was defined as a closed domain; (14) some additional details were added to the experimental methods section; (15) clarification added in the results regarding capillary pressure; (16) uniqueness was further explained; (17) a reference was added for GAM modeling; (18) units were added to pressure; and (19) clarification was on the state function in the conclusions.

We believe we have been responsive to the concerns of this reviewer. In the cases in which the reviewer was wrong, we have explained why this is the case. We understand that a work of this sort covers a lot of ground and is technical, so we made some changes even when we felt they really weren't necessary.

**1 General**

We respond to the comments from Referee #2 beneath comments made. The authors' response is shown as **AU: red**. The changes made are highlighted as **AU: blue**.

**2 Referee #2**

In this manuscript, physically based upscaling of two phase fluid flow in a porous medium is considered by presenting definitions of microscopic and (macroscopic) averaged properties, and investigating this system with experiments and simulations. The manuscript provides nice illustrations of how different experimentally determined pressure differences and local values of capillary pressure are. This is done by a blend of experiments and numerical simulations. While I have no problem with the basic message of the manuscript, the presentation is not as may be expected. Quite some space is reserved for the objectives, a literature overview in the background section, and the presentation of eqs. (1)-(19), which are basically definitions. What remains underexposed, though, is a clear identification of what is new. Certainly, averaging is not, and neither is it for two fluid systems in porous material. Therefore, I propose that this is explicitly mentioned on these sections 2-4.1, as I am not convinced that these sections should be maintained in this manuscript. The aspect of connectivity is given some emphasis (e.g. p.8) and reference is made to McClure et al. Again, I propose that it is clearly identified whether and what is new in this work, as the current text is not clarifying this. Later on, again the experimental and simulation parts appear to be based on work of McClure et al. and it is apparent that this work may duplicate that earlier work. Though the present manuscript is illustrative, I would consider it not fit for publication, if in essence the material is a duplication of earlier work.

**AU: The introduction was written to put this work into a broader**

context associated with the special issue in honor of Professor Wood. Since this may well be the only manuscript on porous media in the issue, it seems some effort should be expended to make the connection with approaches to other hydrological problems. The objectives are brief, and we don't see anything to cut here. The formulation is included because the focus on individual regions within the porous medium is needed to clearly explain the issues involved with disconnected phases.

There are a number of important differences between the McClure et al. manuscript published in *Physical Review E* and this manuscript. Specifically, the most significant differences with the PRE paper are

(1) this manuscript includes data from drainage in addition to randomly initialized configurations allowing a comparison of a new approach with a traditional approach;
(2) this manuscript considers a system where there is experimental support; and
(3) this work focuses on wetting-phase connectivity rather than non-wetting phase connectivity.

Reference to McClure et. al is necessary to provide additional theoretical details on the random phase initialization, but there is not really a significant topical overlap between the two papers. A sentence or two can be inserted to clearly assert what is the new contribution of the current effort.

AU: The abstract has been substantially revised and the contributions from this work are clearly stated up front.

One of the issues that is quite central to this manuscript is that equilibrium is achieved. Considering the small size of the apparatus, I wonder how this is checked.

AU: We are willing to add additional details to the experimental methods section to explain how these data were collected and how we became convinced that equilibrium conditions existed.

AU: We have added these details to the experimental methods.

On several other statements I also wonder what their justification is. Presumably, this is indicated in the cited references, but as a stand-alone manuscript, important statements need to be justified here.

AU: Not sure what statements this comment refers to.

specific comments: 1. I wonder about some of the English (is the term microfluidic well used;

**AU: Yes this is standard terminology that is broadly used. A Google search on "microfluidic" provides over 4M hits (including both microfluidic and microfluidics)**

abstract; these instances on p.4 line 11). The abstract contains quite some text, which I would rate as context, that is not necessary for an abstract and must be deleted: lines 1-8 or even 1-11.

**AU: The abstract could indeed be shortened, but we wanted to ensure, given the special nature of this issue in honor of Eric Wood, that the abstract is sufficiently informative. We would do as the editor wished with regard to this point, but our preference for the reason cited is to leave this longer version pretty much as is.**

**AU: We shortened the abstract substantially and removed material that was not central to the work performed here.**

In addition, the reduction of water content to below the irreducible saturation is mentioned: As the authors make a call for rigorous definitions, I think this contradiction in the text is inappropriate.

**AU: We will make it clear in a revised version that the standard terminology is indeed a misnomer.**

**AU: We added a statement regarding this common misnomer in the computational section and in the abstract added the adjective so-called to alert the reader to this point early on as well.**

Of course, in a special issue focused on Eric Wood, there is a temptation to give some thoughts on his career. However, in this manuscript, those thoughts look quite artificial and unnatural. I would omit those parts of the text.

**AU: The fact that this paper is intended to be part of a special issue in honor of Eric Wood was our motivation for a broad introduction that sought commonalities between Professor Wood's work and this particular, focused piece of work. Although one might argue that the contexts of upscaling in this work and in Prof. Wood's work are different, we assert that the upscaling techniques that can be used are the same. The fact that experimental and computational efforts to support the theoretical results in the two contexts will be different does not detract from the fact that upscaling, in any context, requires support. Again, we are willing to respond to the editor's recommendation on this point, but we prefer to leave this**

portion in tact in light of the nature of the special issue.

AU: We have generally left these comments stand and believe it is appropriate to do so. The connection with our work is much closer than might be apparent without these comments.

2. Averaging (p.4) is older. For instance De Josselin de Jong (around 1955)

AU: We do not know of another source in which the averages computed in this work, and necessary to make the points about the role of connectivity, have been formulated. This material should stay in our view and is necessary for understanding.

AU: This formulation is central to the work, is unique because it is based upon connected components, and has been retained.

3. p.5 line 2: I would add: does not ONLY depend....

AU: Actually, the statement is correct as stands. Capillary pressure is the product of interfacial tension and the mean curvature of the fluid-fluid interface—fluid pressures do not enter the expression. Only at equilibrium is the capillary pressure equal to the pressure difference on either side of the interface. We believe these points are clear in the manuscript.

AU: No changes made.

4. I do understand neither the notation nor the meaning of (2) or the term "extent measure". Please clarify.

AU: We find this comment confusing. Previously, the reviewer opined that this was a standard formulation that wasn't new, and now she/he seems confused about fundamental components of modern averaging theory. We can add some additional references here to help readers who lack the appropriate background but are trying to understand the details of what is written. The indicator function is identified and described in Eqn (2) concisely and correctly. Similarly, the meaning of extent measures is defined right beneath their formulation in Eqn(3).

AU: Slight changes have been made to define the specific interfacial area and specific common curve length.

5. page 8: the term averaged phase pressures is used. I think that it is not appropriately, especially for this manuscript, to be vague about "over what is averaged".

**AU: The purpose of including the formulation is to define precisely every quantity that is used. There is no ambiguity as every symbol is defined completely and in detail, with averages explicitly denoted in terms of their smaller scale precursors.**

**AU: No changes have been made regarding this comment.**

6. in Fig.1, the black circles represent solid phase particles. Are these in fact porous cylinders as I understand from p.9 line 15? I think this info should be made very explicit, to address whether or not this experiment is true 2D or in fact 3D (with additional complications that will be obvious).

**AU: The caption of Figure 1 clearly stated what the reviewer surmised to be the case, including a note of the portion of the domain accessible to fluid flow.**

**AU: No change needed.**

One complication that may not be left undiscussed is that of boundary effects (at front and rear plates). In the same context, I do not understand p.11 line 4-5: why the "depth" (in Fig.1: vertical, horizontal,...) of the real apparatus and of the model differ.

**AU: The two principal radii of curvature are $R_1$ and $R_2$. Since the depth of the micromodel sample is fixed, $R_2$ is fixed, and variations in the mean curvature are due solely to changes in $R_1$. The depth of the simulation domain was increased to improve the numerical accuracy, noting that this approach was sufficient to resolve the behavior of $R_1$.**

**AU: No change made.**

7. Is the instrument new? I ask because it is not clear whether the experiments, their interpretation and such are new and in what sense (see p.10 line 17-18).

**AU: The experimental work reported here is new.**

8. you create random initial conditions below irreducible saturation (p.11). Only now, it is indicated clearly what makes it irreducible: because it is not connected to the wetting phase reservoir. I think that this needs to be mentioned earlier. Also, explain why it is relevant: these situations cannot develop in reality (for the experimental set up) as it is a state below irreducible. You mention (p.11) that below irreducible saturation, where sub-regions are unconnected, this leads to history dependence. I would think that the same is true in the random initial conditions simulations. Where you inject your "blocks", is simulating history.

AU: We agree that the use of the expression "irreducible saturation" is a bad historical misnomer. This name arose from the form of $p^c - s^{\overline{\overline{w}}}$ curves and the experimental methods used to obtain them. In fact, saturations below the "irreducible saturation" exist and can be achieved experimentally. Encouraging abandonment of this unfortunate, yet deeply ingrained, terminology is a huge task. The term, "history dependent" is employed, perhaps in a traditionally inappropriate way, to indicate that the microscale state of the system cannot be characterized by the macroscale variables $p^c$ and $s^{\overline{\overline{w}}}$. If we consider the relationship $p^c(s^{\overline{\overline{w}}})$, we observe "history dependence," as it is traditionally explained, is a consequence of the fact that for a given $s^w$ there are many possible microstates; and all of these do not produce the same value of $p^c$. These microstates can be achieved by operating an experimental apparatus under different scenarios changes of the boundary conditions. However, if we include interfacial area, $\epsilon^{wn}$, in the theoretical construct, then the relationship $p^c(s^{\overline{\overline{w}}}, \epsilon^{wn})$ is able to characterize the possible microstates more effectively, independent of experimental operating strategies and histories, which does remove "history dependence."

AU: As previously noted, we have pointed out in the computational section that irreducible saturation is indeed a misnomer.

[revised manuscript text omitted]

---

## Author Response (AR2)

Second Response to Reviews
**On the Consistency of Scale Among Experiments, Theory, and Simulation**
J.E. McClure, A.L. Dye, W. G. Gray, and C. T. Miller

hess-2016-451

**1 General**

We respond to the comments beneath the comments made. The authors' response is shown as **AU: red**.

**2 Editor Decision**

Editor Decision: Publish subject to revisions (further review by Editor and Referees) (10 Jan 2017) by Prof. Remko Uijlenhoet
Comments to the Author:
Dear authors,

Thanks for submitting a revised version of your manuscript and replies to the reviewers' comments. Both reviewers have evaluated your revised manuscript. Reviewer #2 has some concerns about the presentation style of the manuscript, but finds the contents of the paper generally acceptable. However, reviewer #1 still has serious reservations with your paper, questioning the novelty of the presented results and their relevance to hydrology.

Based on these reviews and my own appreciation of your revised manuscript and rebuttal, I suggest to consider the comments and suggestions provided by both reviewers carefully and provide responses where possible. Also indicate where this could lead to revised formulations in the paper. In particular, I would like to ask you to clarify better where and how this paper adds to previously published work. Thank you very much in advance.

Best regards,

Remko Uijlenhoet

**3 Report 1**

Anonymous during peer-review: Yes
Anonymous in acknowledgements of published article: Yes

Recommendation to the Editor
1) Scientific Significance Does the manuscript represent a substantial contribution to scientific progress within the scope of this journal (substantial new concepts, ideas, methods, or data)? Fair

2) Scientific Quality Are the scientific approach and applied methods valid? Are the results discussed in an appropriate and balanced way (consideration of related work, including appropriate references)? Good

3) Presentation Quality Are the scientific results and conclusions presented in a clear, concise, and well structured way (number and quality of figures/tables, appropriate use of English language)? Fair

For final publication, the manuscript should be rejected

Please note that this rating only refers to this version of the manuscript!

Suggestions for revision or reasons for rejection (will be published if the paper is accepted for final publication)
Not suffiicient new material to warrant publication.
Limited relevance for hydrology.

**3.1 Comments embedded in text of first revised manuscript with highlighted text in quotes.**

Page 4, First two lines of Background Section. "Two spatial scales are of primary interest for the porous medium problems of focus herein: the microscale, which is often referred to as the pore scale; and the macroscale, which is often referred to as the porous medium continuum scale." This helps, as well as the subsequent text. With this terminology, the isuue of relevance remains at what one of the authors in another publication called the megascale, which is what I considered the macroscale before this clarification. I maintain that TCAT is not readily applied at that scale. In very practical terms: I do not see Modflow being replaced anytime soon by a thermodynamically sound model (what is termed 'mature model' in the paper).

**AU: The inability of the reviewer to see that TCAT can be applied at the megascale is his limitation, not a limitation of the method. Papers appear in the literature that are based on megascale TCAT. The authors have applied this method in looking at a full packed column, in stream hydraulics, and for single-phase porous media flow [Gray and Miller, 2009]. Others have tried to apply an averaging theory to watershed analysis [Reggiani & coworkers], although their efforts are limited by failure to properly deal with megascale thermodynamics.**

**We speculate that this comment might be the result of the reviewer not understanding the meaning of megascale, which is the system scale. At the megascale, the details within the domain are not resolved and only the overall behavior of the system as a whole is considered through conditions on the boundary of the domain. Most modeling of porous medium systems is done at the macroscale level, including the MODFLOW code that the reviewer mentions. Thus, our work applies at the traditional scale that porous media models are formulated at.**

**We do not understand the relevance of the comment regarding MODFLOW. This work deals with two-fluid-phase flow in a porous media, and MODFLOW is a production code for solving single-fluid-phase flow. In any event, our goal here is not to produce or apply a production code, rather we endeavor to advance fundamental understanding by pointing out deficiencies in current approaches and demonstrating how these deficiencies might be overcome. Ultimately, we believe that our work can lead to improved production simulators of higher fidelity than current models, but this is down the road and not an objective of the present manuscript.**

Page 6. last line before section 4. "large scale systems". At the megascale.

Page 6. First line of section 4. "An important". This paragraph helps in positioning the paper and stating its purpose.

**AU: We are pleased with the reviewer's reaction.**

Page 10 first line of section 5.1 "To meet the objectives of this work". I still maintain that the Introduction in combination with the objectives create the expectation that you are going to offer something that is directly relevant for two-phase flow at the system scale (that would be unsaturated flow in a field or a landscape, with air and water as the fluids).

**AU: We consider our work to be relevant and applicable to study of a system where air and water are the fluids. The reality is that disconnected regions of wetting phase are common in these systems, which is not in dispute. We have changed the Objectives section to make our objectives clearer. Accounting for these disconnected phase regions is essential to obtain a scale-consistent measure of the phase pressure or capillary pressure. We recognize that one person's assessment of something as "relevant" or "directly relevant" is often different from that of another individual depending on interest in practicality, theory, field work, simulation, code development, or laboratory measurement.**

Page 10 seventh line of section 5.1 "macroscale REV". I have read your response and understand your view better now. You aim at the smallest possible size of the REV, whereas in heterogeneous media there is a range of REV sizes, starting with the size that you defined in your response, and ending at the size where heteogeneities affect the macroscale properties (e.g., Bear, J., and Y. Bachmat, 1991, Introduction to modeling of transport phenomena in porous media, Kluwer, Londdon, p. 24-29). Heterogeneities at that scale are outside the scope of this work. That does not invalidate the study, but it does impose limits on the scope of what it can achieve that are not refelected in the current introduction and the set of objectives.

**AU: We are pleased that the reviewer better understands our situation based on our revisions. Multiphase flow can be complicated by heterogeneity and by disconnection of phases. In fact, fluid phase distributions can be heterogeneous even if the solid is homogeneous. We are looking at this aspect of the problem. However, we note that TCAT applies at any sized macroscale REV where a separation of length scales occurs. Thus the implied limitation of the theory suggested by the reviewer is not accurate. It is just the case that we have focused on a small REV to examine the fundamental issue of phase connectivity for this particular work so that we could perform highly resolved experiments and computations to support our work. Our view is that we want an accurate model for idealized systems that properly resolve the observed physics; then issues at larger scales can be dealt with in turn.**

Page 12 first two lines of text. "The common identification of a saturation as "irreducible" is a misnomer because wetting phase saturations beneath this value can be achieved through, for example, evaporation or by initializing a saturation below this value in an experimental setup." I agree with the sentiment, but I note that Richards' equation implicitly assumes the soil air pressure to be atmospheric at all times and locations, and the water pressure to be continuous. This implicitly requires both phases to be continuous. The fact

that an irreducible water content is assumed to exist by some soil hydraulic parameterizations (among which the most popular ones, regrettably) therefore is a separate issue from the breakdown of the validity of Richards' equation when pockets of isolated water and/or air exist in the soil, or when the soil dries out to such an extent that the water retrieves into pendular rings (which can be considered an extreme case of isolated pockets of water surrounded by a continuous gas phase).

The concept of irreducible water content has been criticized on several occasions in the soil physics literature (no references given because it is tangential to the scope of this paper) because it creates other problems when modeling water movement in dry soils. But is generally considered different from the occurrence of isolated pockets of fluid, which even challenges the underlying differential equation.

**AU: The issues of disconnected phases and irreducible saturation is central to this manuscript. The reviewer points to Richards' equations and typical closure relations and the implicit assumptions as regrettable. This is the problem we are focused on. Our work looks at disconnected fluid phases and examines what is commonly referred to as irreducible saturation. There are many misnomers in hydrology. We are not so much interested in advocating for proper usage of language in this particular paper. We are advocating for proper representation of processes.**

Page 13. Section 5.3. line 6. "irreducible wetting phase saturation." I agree it can play an important role if the simulations enter into very dry territory, which they usuall didn't at the time these parameterizations were developed. Aside from that, I do not consider this an artifact of the experimental design, but of the mathematical formulation of the relationship between capillary pressure and fluid saturation.

**AU: Our view is that evapotranspiration routinely lowers moisture content beneath the value referred to as irreducible saturation. Typical closure relations do not handle this situation. We also consider the more general case of disconnected phases, which also routinely occur in real systems. Indeed, it is difficult to saturate even a porous medium packed column in the laboratory without entrapping a gas phase. We show that these common cases can be modeled accurately with a closure relation based upon general additive models thus overcoming the shortcoming of traditional approaches to closure. Thus, we have developed a more complete representation of the dependence of capillary pressure on other variables of the system.**

Page 14, line 3 "by analyzing the curvature of the fluid-fluid interface." This

points to an experimental limitation, does it not? Observation in the field will be impossible, and lab observations require conditions that permit an inspection of these interfaces by whatever means.

**AU: This paper makes the point that curvature can be observed in micromodels, in LB models and in three-dimensional porous media using experimental micro-computed tomography ($\mu$CT). The capillary pressure, which relates to the curvature, is an important parameter. Although it may not be observable in some situations, we must account for its effects. New experimental data sources such as $\mu$CT are already used to observe previously inaccessible information such as interfacial curvatures. By looking at the kinds of systems we are studying, we gain insight into this parameter and how it must be incorporated into models if the physics are to be accounted for. Saturation is inadequate as a single parameter to account for curvature precisely because of the way phases distribute within the system. We are interested in building a sound theoretical framework that makes use of operative physics. If properties of operative physics are difficult to observe, that does not mean one is at liberty to neglect this element of physics.**

**4  Report 2**

Anonymous during peer-review: Yes
Anonymous in acknowledgements of published article: Yes

Recommendation to the Editor
1) Scientific Significance Does the manuscript represent a substantial contribution to scientific progress within the scope of this journal (substantial new concepts, ideas, methods, or data)? Good

2) Scientific Quality Are the scientific approach and applied methods valid? Are the results discussed in an appropriate and balanced way (consideration of related work, including appropriate references)? Good

3) Presentation Quality Are the scientific results and conclusions presented in a clear, concise, and well structured way (number and quality of figures/tables, appropriate use of English language)? Fair

For final publication, the manuscript should be accepted subject to minor revisions

Please note that this rating only refers to this version of the manuscript!

Suggestions for revision or reasons for rejection (will be published if the paper is accepted for final publication) I have mentioned earlier on the extensive objectives, background, and quite basic eqs (1-19), which I think are basically definitions. The authors refer to the broader context in honor of Prof Wood, that their manuscript may be the only one on porous media, and therefore reasonable to connect with other approaches.

**AU: We apparently disagree with this reviewer on the importance of equations (1)–(19). We believe very strongly that quantities have not been carefully defined in porous media studies. The term "pressure" is tossed around without defining it at a scale or in such a way that one must know how to measure it. In fact the change in scale may be the culprit for this. Since our point is to ensure that quantities may be compared effectively among theory, experiment, and simulation only if the quantities discussed are defined in the same way, it is important to make a point of explicitly defining the various quantities. There are several measures of pressure described, and it is eminently worthwhile to understand the distinctions. We thus find this important and central to the point of the manuscript. This formulation is also not available elsewhere as the reviewer suggests, because it is formulated for the general case of disconnected regions.**

This reviewer has of course no objection that Prof. Wood is honored by a special issue (presumably with an editorial to highlight Prof. Wood's great achievements). But this manuscript is a normal paper, not an editorial and not a review. Therefore, I read it as a paper and I find this connection far from convincing. Instead, to me it reads as an artificial attempt to broaden the perspective of this paper. That this paper may be the only one on porous media is completely irrelevant, as I see it. The manuscript is not a chapter of a book that intends completeness to some degree, I see no reason why the special issue should pretend that. Again: this is a normal paper, and such papers should be to the point. I can extend this point to the abstract. An abstract is a short description of the main points of the manuscript, with primary findings. A short editorial on Prof. Wood (though sympathetic) as well as insightful comments that are not basic to this paper should be omitted. Though the response considers that upscaling techniques are the same, whatever scale and context, maybe appropriate for a review article, but is beside the point in a research article, for the same reason as the response argues for being to the point in its response of averaging references: "...the averages computed in this work". I have indicated the lines that I think are obsolete in the abstract of this submission in the first round.

**AU: With all due respect, we think our abstract highlights what we want to accomplish; and we did rewrite the abstract based upon this reviewer's original comments. Perhaps this was missed in the re-review. Regarding issues related to Prof. Wood and the broader context, we understand well the reviewer's perspective, and it also has merit. Our paper is indeed a "normal" paper. The fact that it appears in a collection intended to honor Prof. Wood actually opens a door to make points about averaging, scale, thermodynamics, capillary pressure, etc. that are not normally considered or encountered by the bulk of the community that works in the same area as Prof. Wood. We thus want to take advantage of an opportunity to make the methods we use known to a broader segment of the HESS community and to others involved in modeling. One could argue, as the reviewer essentially does, that the prospective audience should not be considered. We simply disagree.**

I also still find that six bullets in the objectives section are overdoing it. To me it looks as if every breath has to be announced. I would suggest that they are following the suggestion of the author himself regarding possible overlap with the much cited McClure et al paper: "A sentence or two can be inserted to clearly assert what is the new contribution...". Please go ahead (despite that you need three points to highlight the most significant differences with the PRE paper) and do the same with the objectives.

**AU: We have rewritten the Objectives section.**

On my remark on how it is checked that equilibrium is reached, the author states the willingness to add an apparently necessary experimental methods section (if really needed!). I find it quite remarkable, that to convince the readership of this equilibrium, apparently an entire section is needed, that apparently was considered obsolete in the first submission, and apparently is not yet sufficiently described in the McClure et al. or other papers. I am also somewhat confused by this response: is the content of such a section not part of another paper (hence, should be presented here), why is it then still not needed, and is the provided statement on this issue in the response sufficient or not?

**AU: There appears to be some confusion here. We have made a minor addition to the experimental methods section present in the original manuscript that explains how we knew we were at an equilibrium state. This would seem to have responded to this concern with, as the reviewer suggests, a small addition. We believe our original response document was partially responsible for this confusion, but indeed the revised manuscript as it stands has addressed this point adequately in our view.**

In my earlier comments, I mentioned that the equations in section 4.1 are basically definitions. In response, I get the pun that apparently I lack the appropriate background. I would think that such a set of definitions can be found in handbooks, other articles and are therefore obsolete in this manuscript, as they are definitions. For this manuscript, this section can be shortened to what matters. In that case, it may be less painful, if the authors use notations of cited or not cited references, without explaining it.

**AU: This comment was raised above and answered there.**

The response on Figure 1 is fine, but I suggest that for instance the caption explicitly mentions the black circles to be pure solid.

**AU: We have made this change.**

It is still not completely clear to me, if R2 in the depth direction is 2.2 micrometer, and the black circles differ in size (horizontal plane): these black volumes are therefore not pure spheres? You might mention so, as Figure 1 could be interpreted differently.

**AU: We have clarified these points in the revised manuscript. The solids are cylinders that are void of openings.**

I am glad to hear that the experimental work is new. However, I asked about the instrument also.

**AU: The micromodel cell is new. The other microfluidic methods have been previously used.**

Thank you for your response to issue 8. I understand from your answer, that if interfacial area is included, all possible microstates collapse into one relationship, thus eliminating 'history'.

**AU: This is close to true. Actually, the Euler characteristic is also needed for the highest fidelity representation.**

[revised manuscript text omitted]

---

## Author Response (AR3)

Third Response to Reviews
**On the Consistency of Scale Among Experiments, Theory, and Simulation**
J.E. McClure, A.L. Dye, W. G. Gray, and C. T. Miller

hess-2016-451

**1   General**

We respond to the comments beneath the comments made. The authors' response is shown as **AU: red**.

**2   Editor Decision**

Editor Decision: Publish subject to revisions (further review by Editor and Referees) (10 Jan 2017) by Prof. Remko Uijlenhoet
Comments to the Author:
Dear authors,

Thank you very much for submitting your replies to the issues raised by the reviewers and for submitting the revised version of your paper.

As you can see, reviewer #1 still feels the paper will probably not easily fit into the scope of the other papers to be published as part of the Special Issue of HESS honoring Prof. Eric Wood. The main point of this reviewer is that the spatial scale at which you deal with the topic of scale consistency among experiments, theory and simulation (namely that of soil as a porous medium) is likely going to be quite disparate from the spatial scales at which Prof. Wood and many of his colleagues have been and are working (namely that of landscapes, catchments, river basins, and continents). The reviewer is therefore not convinced that your work is going to provide new perspectives for hydrologists trying to model the world at such larger scales. That may be true, but at the same time the reviewer does not seem to have major concerns with the technical details of your work. Therefore, I am inclined to leave the discussion about the value of your work from a broader hydrological perspective to the readers of the pages of HESS and other scientific journals rather than to the reviewers of your manuscript alone.

**AU: We are of the opinion that Reviewer #1 has a particular area of hydrology that is of his/her interest. This area seems to be more**

**applied than the subject of our manuscript. This difference in emphasis does not mean that we must conform our paper to the interests of the reviewer. Nor does it mean that the reviewer must conform his/her interests to those of the authors. The issues of importance should be related to the technical aspects of the paper. This reviewer, or any reviewer, is handicapped when the topic of the paper does not fall into his/her area of expertise. From our perspective, the technical statements of the reviewer have been dealt with appropriately. Hopefully, this has added some clarity to the work.**

Reviewer #2 also seems to have no major issues with the technical content of your revised manuscript. This reviewer still finds the way in which you present your work, including your reference to Prof. Wood's work, not appropriate for a regular paper that is submitted to become part of a special issue. Although I am sympathetic with this reviewer's perspective, I feel that the presentation style is merely a matter of taste.

**AU: The reviewer has not updated his/her comments, but we have received your clarification from this reviewer. We agree that there is a stylistic difference between what we have done and what the reviewer suggests. We prefer the style we have chosen and find it satisfactory. We have made a few changes to try to accommodate the reviewer while maintaining our preferences for our paper.**

In conclusion, I recommend to accept your manuscript subject to minor revisions. As far as I am concerned, the paper will not be sent out for review again. I would appreciate it very much if you could give it one last try to accommodate some of the issues raised by the referees. Thank you very much in advance. I look forward to handling the revised version of your manuscript.

**AU: This has been a most tedious review process. We appreciate the efforts of those who looked at this paper to improve the manuscript. It can be difficult for reviewers when the vision of authors is different from theirs. Nonetheless, the reviewers have given us an opportunity to consider a different perspective and make sure that our work is presented as we would like. We are grateful for having had that opportunity.**

Best regards,

Remko Uijlenhoet

**3    Report 1**

Anonymous during peer-review: Yes
Anonymous in acknowledgements of published article: Yes

Recommendation to the Editor
1) Scientific Significance Does the manuscript represent a substantial contribution to scientific progress within the scope of this journal (substantial new concepts, ideas, methods, or data)? Fair

2) Scientific Quality
Are the scientific approach and applied methods valid? Are the results discussed in an appropriate and balanced way (consideration of related work, including appropriate references)? Good

3) Presentation Quality
Are the scientific results and conclusions presented in a clear, concise, and well structured way (number and quality of figures/tables, appropriate use of English language)? Good

For final publication, the manuscript should be rejected

Please note that this rating only refers to this version of the manuscript!

Suggestions for revision or reasons for rejection (will be published if the paper is accepted for final publication)
Not sufficient new material to warrant publication.
Limited relevance for hydrology.

**3.1   Comments embedded in text of first revised manuscript with highlighted text in quotes.**

No significant changes were made. I read the replies and cannot escape the impression that in the communication between the authors and this reviewer both sides appear to have some difficulties to grasp what the other side means to convey. I have made progress there and gained a better understanding, but the rather tort response from the authors shows that this is not mutual. As a case in point, my reference to MODFLOW was an attempt to try to get one of my points across a bit differently, and certainly not an invitation to build a TCAT version of it. A full paragraph discussing the limitations of MODFLOW would not have been necessary.

**AU: We have done our best to address the reviewer's comments as they were presented and as we understand them. The two subjec-**

tive comments remaining for this reviewer are that the manuscript does not contain sufficient new material to warrant publication and that the work presented is not sufficiently relevant to field-scale hydrology—perhaps these two items are related in the view of the reviewer.

Resolution of these issues depends upon one's vision of the role and importance of fundamental research in an applied field. Put another way, do we accept current model formulations and do our best with them in light of difficult application issues, or do we attempt to identify ways in which current models can better represent the systems that we wish to approximate? Our view is that we need to do both of these things, while the reviewer evidently believes improving current models is either unimportant or a hopeless task. This is an interesting philosophical point. We feel sure that both the authors and the reviewer are interested in improving our ability to describe complex hydrologic systems. The avenue toward the most important and rapid advancements is often not clear. We favor a climate of mutual intellectual respect in which the ideas of all working toward the resolution of societal problems are encouraged to contribute, their work reviewed for scientific accuracy, and the impact to the field revealed in time by the community and those that build upon the work.

We believe that our view point is not unique in regard to the potential value of the small-scale approaches taken in this work. To support this claim, we performed a Google search on pore scale modeling of porous media, and this search returned about 870,000 results. Special issues dedicated to pore scale modeling have occurred routinely in the hydrologic literature, and special sessions on this topic convened at virtually every hydrologic venue with which we are familiar. Our conclusion is that others have found some worth in the sorts of approaches taken in this work, and all scientifically valid avenues toward potential advancements are worthy of consideration.

However, to address this point, we have added a paragraph at the end of the objectives to explain that this work addresses a focused fundamental issue using experimental and computational approaches at a scale that is much smaller than field-scale hydrologic problems and noting that this work is ideally a stepping stone toward improved understanding and models of hydrologic systems.

I have some background in the area the authors are covering, but after providing extensive detailed comments to help the authors identify parts of the

paper that are hard to follow or otherwise would benefit from modification am being informed that my limitations undermine my ability to understand the contribution of the paper. I am rather pessimistic in fearing that the vast majority of the HESS readership will be burdened by similar if not larger limitations.

**AU: We do not share the reviewer's pessimistic opinion of the abilities of the HESS readership.**

To see the contribution of the paper it helps to be aware that most hydrological problems manifest themselves and need solutions at the megascale. Hydrology is moving in the direction of modeling megascale problems with macroscale models through the use of brutal computing power for running models with a very large number of nodes and sophisticated parameter optimization techniques. In the revision the authors do a better job of highlighting the limitations of these models, but most hydrologists are aware of that, although usually not with the thermodynamical rigour that I ascribe to the authors.

At the megascale, hydrologsts face massive issues with heterogeneity, spatial and temporal dynamics in porous medium properties caused by such factors as swelling and shrinkage, bioturbation, tillage, land use changes, and what not. The scientifically elegant and physically imperative approach advocated in this work is not generally considered the biggest fish to fry.

For the work to make a significant contribution, some expansion is still required, as per my earlier suggestions and those put forward by the authors.

**AU: These last three paragraphs boil down the areas of contention between the reviewer and the authors. We do not disagree at all with the reviewer's statements concerning the need for megascale solutions of hydrologic problems, of the need to harness computing power, of the importance of heterogeneity, swelling, bioturbation, tillage, land use changes, and subsidence. We admire the work of many individuals who address these problems in a timely manner as solution of these problems is crucial for the welfare of society. Applied hydrologic research is extremely important and worthwhile!**

**At the same time, we adopt the perspective that using traditional theoretical descriptions of hydrologic problems in an era where computing power is exploding, techniques for investigating sub-pore-scale processes are developing, and where the ability to manage and manipulate data is unprecedented overlooks an important element of hydrologic studies. Giants of the past did the best they could in describing problem physics in a way that would allow these descriptions to be used to simulate or describe system behavior. Few**

would claim that this work has been overwhelmingly successful. We believe that by ensuring that the fundamental physical description of hydrologic problems takes advantage of new methods of model support, better megascale models can and will result. We know that the work in the present paper does not immediately "plug in" to megascale models and reduce the difficulty that megascale modelers are currently wrestling with. We do claim that this systematic approach to dealing with changes of scale can be a boon to the ultimate development of megascale models that bury some elements of the system physics in fitting parameters. If one is in a hurry to get a megascale result, any result, then this work is not a significant contribution as the reviewer suggests. If, on the other hand, one values fundamental work that has the potential to support more elegant and complete field-scale models that capture operative mechanisms, this work is a contribution whose significance will be determined in time as this next generation of models develops. For example, the use of hysteresis and irreducible saturation in current models is an artifact of the failure to account for small scale physics. We now have ways to formulate the physics better based on fundamental considerations. Work is needed to include these advancements in production codes. We also maintain that the consistent method of scale change employed here in dealing with microscale and macroscale variables is a method that can be employed in subsequent transfers of scale to the megascale while ensuring consistent definition of variables. This is in contrast to some models currently being advocated that do not transfer scales carefully and thus do not provide a means of relating measurements one might make in the field to information developed in a laboratory or in a theoretical model.

**4  Report 2**

Anonymous during peer-review: Yes
Anonymous in acknowledgements of published article: Yes

Recommendation to the Editor
1) Scientific Significance Does the manuscript represent a substantial contribution to scientific progress within the scope of this journal (substantial new concepts, ideas, methods, or data)? Good

2) Scientific Quality
Are the scientific approach and applied methods valid? Are the results discussed in an appropriate and balanced way (consideration of related work,

including appropriate references)? Good

3) Presentation Quality Are the scientific results and conclusions presented in a clear, concise, and well structured way (number and quality of figures/tables, appropriate use of English language)? Fair

For final publication, the manuscript should be accepted subject to minor revisions

Please note that this rating only refers to this version of the manuscript!

Suggestions for revision or reasons for rejection (will be published if the paper is accepted for final publication)

**AU: The editor has noted that this reviewer has no major issues with the technical content of the revised manuscript. The reviewer did not provide an updated set of comments, but included comments we previously considered and answered as best we could. We have added yet another round of notes and comments trying to allay the discomfort that this reviewer has with our chosen method of presentation.**

**The editor did provide some additional guidance through email regarding this reviewer's comments, which centered on relation of this work to Professor Wood's work and evidently on the objectives. In response to these comments, we have deleted reference to Professor Wood from the abstract. Also note that we revised and shortened the objectives in the last version; and we are confident that these objectives are succinct, accurate, and met in this manuscript.**

**This reviewer seems unhappy with the listing of equations that define variables. These different definitions are important and identify precisely what we are measuring, comparing, and manipulating. The word "pressure" is used ubiquitously in studies of porous media. However, the actual meaning of the word in relation to data measured or modeled is, at best, glossed over. Our work requires and utilizes clear and careful definitions. Thus we believe the equations are a very important part of this work. We have added the second paragraph in the theory section which explains why we think it is necessary to clearly define larger scale variables in terms of microscale precursors in equations (1) – (19).**

**We have gone through the remainder of the paper, as well, but have**

made no other significant changes.